# Alzheimer's disease plasma biomarkers are associated with cognitive performance among Hispanic/Latino adults

## Abstract

**Background** Blood-based biomarkers hold promise as a minimally invasive tool for identifying early signs of Alzheimer's disease pathology and neurodegeneration. We investigated associations between plasma biomarkers of amyloid-beta, tau, neuroaxonal injury, and glial activation with cognitive performance among community-dwelling Hispanic/Latino adults in the United States.

**Methods** We analyzed cross-sectional data from 5730 adults aged 50 years and older (unweighted; mean [SD], 63.5 [8.2] years) in the *Study of Latinos-Investigation of Neurocognitive Aging* (SOL-INCA; 2016–2018). Plasma concentrations of amyloid-beta ($A\beta_{42/40}$), phosphorylated tau-181 (pTau-181), neurofilament light chain (NfL), and glial fibrillary acidic protein (GFAP) were quantified (Quanterix Simoa HD-X) and log-transformed (ln). Cognitive performance was assessed across domain-specific scores (learning, memory, verbal fluency, and executive functioning/processing speed) used to calculate global cognitive performance. Survey-weighted linear regression models were used to examine associations between plasma biomarkers and cognitive performance, adjusting for sociodemographic, cardiometabolic, kidney, and APOE ε4 covariates.

**Results** Here we show higher ln(pTau-181) and ln(NfL) are associated with lower global cognitive performance ($b_{pTau-181} = -0.06$; 95%CI = [−0.12;−0.01]; $p = 0.022$; $b_{NfL} = -0.07$; 95%CI = [−0.12;−0.02]; $p = 0.005$). Lower ln($A\beta_{42/40}$) is associated with poorer verbal fluency, higher ln(pTau-181) is associated with poorer learning and memory, and higher ln(NfL) is associated with learning and executive functioning/processing speed. We find ln(GFAP) is not significantly associated with cognitive performance.

**Conclusions** Plasma biomarkers related to Alzheimer's disease pathophysiology and broader neurodegenerative processes are associated with cognitive performance among Hispanic/Latino adults. These findings highlight the potential utility of blood-based biomarkers for identifying early cognitive vulnerability in this population.

## Plain language summary

This study looked at how certain substances in the blood, called biomarkers, are linked to thinking and memory abilities in Hispanic/Latino adults aged 50 and older in the United States. These biomarkers can reflect changes in the brain related to Alzheimer's disease and other conditions that affect cognition. Researchers tested blood samples from over 5700 adults aged 50 and older and compared the results with their performance on memory and thinking tests. They found that people with higher levels of certain biomarkers had lower cognitive scores. This research helps us understand how blood tests might one day help detect early brain changes in diverse communities. It also highlights the importance of including underrepresented populations in aging and brain health research.

Blood-based biomarkers (BBMs) have emerged as a promising, minimally invasive, scalable alternative for detecting Alzheimer's disease (AD) pathophysiology in vivo. These include plasma amyloid-beta ($A\beta_{42/40}$), and phosphorylated-tau (pTau-181), both considered core biomarkers of AD neuropathology. Other plasma biomarkers such as neurofilament light chain (NfL), and glial fibrillary acidic protein (GFAP) are not specific to AD but reflect broader neurodegenerative processes, including neuroaxonal damage and neuroinflammation, respectively. These plasma biomarkers are associated with AD and related dementias (ADRD), as well as other aging-related brain conditions[1–3]. Increasing evidence shows that these plasma biomarkers correlate with cerebrospinal fluid (CSF) and neuroimaging measures of AD, and

✉ e-mail: frmarquez@ucsd.edu

increasingly used in both research and clinical settings for early detection, screening, and risk stratification of AD and ADRD[2,4-7].

Despite these advances, there is limited research on plasma biomarkers and cognition among Hispanic/Latino individuals. Hispanic/Latino adults, currently the largest ethnic minority in the United States (US), are projected to experience the largest proportional increase in ADRD in the coming decades[8,9]. Evaluating biomarker-cognition associations in this population is essential, as known ADRD risk factors include age, sex, health factors, and APOE ε4 genotype, and plasma biomarker levels may be influenced by unique cardiometabolic profiles, higher prevalence of chronic kidney disease (CKD), and differences in genetic risk such as ancestry-specific APOE-ε4 allele distribution[4,10-12]. For example, Hispanic/Latino adults experience high rates of obesity, cardiovascular disease risk factors, and CKD, all of which may influence plasma biomarker levels[13-17]. Appropriate use recommendations for BBMs in AD highlight the need to evaluate these biomarkers in varied settings and populations[2,18].

Previous studies examining plasma biomarkers in Hispanic/Latino populations have used small or clinic-based samples, often with specific Hispanic/Latino backgrounds or heritage groups (e.g., Caribbean Hispanic, Mexican American)[19]. The available evidence suggesting that BBM concentrations varies by ethnicity is mixed with one study reporting differences across ethnic and racial groups[20], and others reporting no differences[21-23]. Additionally, few studies have examined these biomarkers in middle-aged individuals, a critical period for intervention. Notably, the association between BBMs and cognitive function has not been described in a large, heterogeneous, community-based sample of Hispanic/Latino adults.

In the present study, we examine associations between plasma biomarkers reflecting amyloid, tau, neurodegeneration, and neuroinflammation and global and domain-specific cognitive performance. Informed by previous research[24,25], we hypothesize that higher plasma biomarker levels of pTau-181 (AD-specific), NfL and GFAP (non-specific ADRD-related), but lower levels of $A\beta_{42/40}$ (AD-specific), are associated with lower cognitive performance. We also sought to investigate whether these associations are consistent when considering age, cognitive status, cardiometabolic risk, kidney dysfunction, and APOE ε4 genotype that they may influence biomarker expression and cognitive performance in this population. In a large, heterogeneous, community-based sample of Hispanic/Latino adults aged 50 and older living in the United States, lower $A\beta_{42/40}$ and higher plasma pTau-181, and NfL levels are associated with poorer cognitive performance, whereas GFAP shows no association. These associations remain after accounting for age, cardiometabolic risk, kidney dysfunction, and APOE ε4 genotype, and they vary by age group and cognitive status. Overall, these findings highlight the potential utility of plasma biomarkers for identifying ADRD-related cognitive risk and population-level screening in this population.

## Methods
### Study population
We analyzed data from the Study of Latinos-Investigation of Neurocognitive Aging (SOL-INCA; 2016–2018), an ancillary study to the Hispanic Community Health Study/Study of Latinos (HCHS/SOL; 2008-2011), a multisite cohort of community-dwelling, self-identified Hispanic/Latino adults sampled from four major U.S. metropolitan areas: Bronx, NY; Chicago, IL; Miami, FL; and San Diego, CA. For this analysis, we included adults aged ≥50 years old ($N = 6377$). Sampling weights accounted for the HCHS/SOL study design and non-response to ensure generalizability to the target population. Details of the HCHS/SOL and SOL-INCA designs and sampling methods have been previously described[26-29]. The HCHS/SOL and the SOL-INCA studies were reviewed and approved by the Institutional Review Boards of the University of California San Diego (Study #803924) and all participating sites (University of North Carolina-Chapel Hill Single IRB: #20-1900), and participants gave written informed consent.

## Neuropsychological assessment and cognitive status
Cognitive testing was administered at the field centers by bicultural/bilingual staff using a brief battery and four cognitive scores were derived: Brief-Spanish English Verbal Learning (B-SEVLT) -Sum (verbal episodic learning; the summed total of correctly learned items across 3 trials; range, 0–45), and -Recall (verbal episodic memory; total correctly recalled items after an interference trial; range, 0–15)[30], Word Fluency (WF; phonemic verbal fluency; sum of correctly generated words within 1 min for the letters F and A), and Digit Symbol Subtest (DSS) score (executive functioning/processing speed; number of correct matches in 90 s)[31]. Raw scores from each test were standardized into z-scores using the mean and standard deviation (SD) of the SOL-INCA target population, and a global cognitive composite score (global cognition; GC) was then derived by averaging the z-scores of the four domain-specific tests. Higher scores indicate better cognitive performance. Additional details on the cognitive battery have been previously published[32].

Mild Cognitive Impairment (MCI) was defined using criteria developed for the SOL-INCA study using National Institutes of Aging – Alzheimer's Association guidelines[33]. Classification was based on neuropsychological test performance in the impaired range (at or below −1 SD below the mean) relative to SOL-INCA internal norms adjusting for age, sex, education, and Picture Vocabulary Test; evidence of global cognitive decline from baseline equal to or exceeding −0.055 SDs per year; self-reported cognitive decline on the brief Everyday Cognition Scale (ECog-12); no or minimal functional impairment in instrumental activities of daily living. Participants with suspected severe cognitive impairment were excluded.

## Plasma biomarker measurement
Fasting blood samples were collected following standardized procedures[28,34]. Plasma aliquots were stored at −80 °C and shipped to the Advanced Research and Diagnostic Laboratory at the University of Minnesota. Aβ40, Aβ42, NfL, and GFAP were assayed with the Simoa Neurology 4-Plex E Advantage kit (Cat. #: 103670). PTau-181 was measured with the Simoa pTau-181 Advantage v2 kit (Cat #: 103714), and a subset of samples were assayed with the Simoa NF-light Advantage kit (Cat #: 103186) for NfL. All samples were assayed using the ultra-sensitive Simoa (single molecule array) technology platform HD-X (Quanterix.com)[35]. $A\beta_{42/40}$ ratios were computed from the Aβ40 and Aβ42 values.

All plasma biomarker measurements were conducted using distinct plasma samples. No repeated measures of the same sample were included in the final analysis. A set of blinded duplicate samples and pooled controls were included for quality control and assay precision assessment but were not part of the analytic dataset. Blinded duplicated showed coefficients of variation (CV) < 11.3%. We included pooled control samples in every assay plate for GFAP, NfL, and pTau-181. These controls, derived from a pool of 30 anonymized donors, demonstrated CVs below 12% across plates. For Aβ40 and Aβ42, we relied on the manufacturer-provided synthetic controls and blinded duplicate samples to monitor precision for these two analytes. Of the 6377 SOL-INCA participants aged ≥50 years, $n = 151$ participants were excluded due to unavailable or unprocessed samples, resulting in 6226 participants eligible for biomarker analysis. Samples with concentrations outside the quantification limits were excluded (Supplementary Fig. 1).

## Covariates
Covariates were selected a priori based on literature and established relevance in HCHS/SOL and SOL-INCA[15,16,36-41]. Descriptive analyses also supported their associations with both plasma biomarkers and cognitive performance in this cohort. Variables included age (in years), sex (female, male), level of education (less than high school, high school or equivalent, more than high school), Hispanic/Latino background (Dominican, Central American, Cuban, Mexican, Puerto Rican, South American, more than one/other), and field center. Cardiometabolic risk factors were defined as follows: body mass index per WHO criteria (BMI in kg/m²; underweight [<18.5], normal [18.5–24.9], overweight [25.0–29.9], obese [30+]), diabetes status defined per ADA criteria (normal glucose regulation, impaired glucose

tolerance, diabetes)[42], hypertension status per NHANES criteria (140/90 mmHg or if they self-reported current antihypertensive mediation use; individuals declining a blood pressure measurement and not reporting medication use were assumed to be not meeting criteria)[43], dyslipidemia status (LDL-cholesterol ≥ 160 mg/dL, or HDL-cholesterol < 40 mg/dL, or triglyceride ≥ 200 mg/dL), and CKD status based on estimated glomerular filtration rate (eGFR; CKD EPI creatinine cystatin C formula[44]) and urine albumin-creatinine ratio (uACR), defined as meeting either condition of eGFR <60 mL/min/1.73m$^2$ or uACR ≥ 30 mg/g. APOE ε4 genotype was also included (presence of one or more ε4 alleles). Single nucleotide polymorphisms *rs429358* and *rs7412*, which define APOE isoforms, were genotyped using TaqMan assays. PCR amplification and genotype calling were performed using standard real-time PCR methods[39]. The covariates were measured at Visit 2 (the SOL-INCA visit), except for sex, Hispanic/Latino background, and field center at Visit 1. Sex was self-reported, and gender identity was not collected. Participants missing covariate data ($n = 496$) were excluded, yielding an analytic sample of 5730 (Supplementary Fig. 1).

### Statistics and reproducibility

We described the study target population (unweighted $N = 5730$) using survey-weighted means and proportions, with differences across age groups assessed using adjusted $F$-tests and chi-square tests (Table 1; Supplementary Table 1). Weighted Pearson's correlation coefficients ($r$) were computed to examine bivariate associations between log-transformed plasma biomarkers and cognitive performance (Supplementary Fig. 2). We also used survey-weighted linear regression models to assess associations between basic demographic characteristics (age, sex, education, Hispanic/Latino background) and cognitive performance in Supplementary Data 1.

We used survey-weighted linear regression models to examine cross-sectional associations between plasma biomarker concentrations and global and domain-specific cognitive performance. Plasma biomarker concentrations were natural log-transformed (ln) to reduce skew. The primary outcome was cognitive performance, expressed as a global composite (average z-scores) and test-specific z-scores. Regression coefficients (unstandardized) and 95% CIs were reported. Six nested models were specified to examine covariate effects: (M0) unadjusted, (M1) adjusted for age, (M2) additionally adjusted for sociodemographic factors, including sex, education, Hispanic/Latino background, and field center, (M3) additionally adjusted for health factors, including BMI, diabetes, hypertension, and dyslipidemia, (M4) additionally adjusted for CKD status, and (M5) additionally adjusted for APOE ε4 genotype (Supplementary Data 3). Forest plots display model estimates and 95% CIs (Fig. 1).

All analyses were conducted using Stata 17 (StataCorp), applying survey (svy) commands to account for HCHS/SOL's complex sampling design, including clustering, stratification and weights. Notably, some regression assumptions (e.g., independent and identically distributed observations) do not hold when using complex survey designs (i.e., clustering, stratification); however, survey-weighted linear regression (e.g., using svy functionalities in Stata) that incorporate the complex sampling design account for these issues. Figures were generated with R (v4.4.2).

All tests were two-sided, and $p < 0.05$ were considered statistically significant. While the primary analyses were hypothesis-driven and focused on a limited number of pre-specified biomarkers and cognitive outcomes, we also applied Benjamini-Hochberg false discovery rate (FDR) correction within cognitive domains as a sensitivity analysis to account for multiple testing[45]. Uncorrected $p$ values and Benjamini-Hochberg FDR-adjusted values are presented in Supplementary Table 2.

### Subgroup/Stratified Analysis

Stratified analyses were conducted by age group (<60; 60–69; 70+; Supplementary Data 3; Fig. 2). Age-stratified models did not adjust for age within each age strata since our interest was in comparing the average associations between exposures and outcomes across these pre-defined age groups. To ensure that our results were not driven by MCI individuals and minimize potential sources of bias, we stratified analyses by cognitive status

(Supplementary Data 5; Fig. 3). Additionally, we fit models (M6) wherein all plasma biomarkers were entered simultaneously while adjusting for the full covariates specified above (adjusted for the same covariates as M5) to assess their joint associations with cognitive performance (Supplementary Data 6; Supplementary Figs. 6, 7).

### Sensitivity analyses

We examined the consistency of our findings using non-log-transformed plasma biomarkers, excluding participants with biomarker concentrations that exceeded or were below 3 SD from the mean: Aβ$_{42/40}$ ($n = 0$), pTau-181 ($n = 56$), NfL ($n = 65$), and GFAP ($n = 3$). The plasma biomarker values were standardized (z-scored) prior to modeling to facilitate interpretation of results in terms of SD changes in the exposures (Supplementary Data 7). We also performed a sensitivity analysis limited to participants with NfL measured using the multiplex assay ($N = 5562$) to ensure stability of results independent of assay (Supplementary Table 3).

## Results

### Target population characteristics

Among 5730 Hispanic/Latino adults (mean [SD] age = 63.5 [8.2] years, 54% female), education, Hispanic/Latino background, diabetes, hypertension, dyslipidemia, and CKD status varied by age group, while sex, BMI category, or APOE ε4 genotype did not (Table 1). Descriptive statistics for plasma biomarkers and cognitive test scores are reported in Supplementary Table 1. Based on SOL-INCA criteria, 9.7% of the analytical sample met criteria for MCI.

### Distributions and correlations of plasma biomarkers

A heat map of the survey-weighted correlations between the natural log-transformed plasma biomarkers and cognitive outcomes are presented in Supplementary Fig. 2, and the scatter plots and linear fit reflecting their associations are presented in Supplementary Fig. 3. ln(pTau-181), ln(NfL), and ln(GFAP) were negatively correlated with global cognition (Pearson's $r = -0.16$, Pearson's $r = -0.24$, and Pearson's $r = -0.18$ respectively), whereas ln(Aβ42/40) was weakly positively correlated (Pearson's $r = 0.07$). Scatter plots with linear fit are provided, including stratified by age group and cognitive status, in Supplementary Figs. 4, 5.

### Associations with global cognitive performance

In fully adjusted models (M5), higher ln(pTau-181) (GC$_{pTau-181}$: $b = -0.06$; 95% CI = [−0.12; −0.01]; $p = 0.022$) and ln(NfL) (GC$_{NfL}$: $b = -0.07$; 95% CI = [−0.12; −0.02]; $p = 0.005$) were associated with worse global cognition. ln(Aβ$_{42/40}$) or ln(GFAP) were not significantly associated with global cognition (Fig. 1, Supplementary Data 2).

### Associations with domain-specific cognitive performance

In fully adjusted models, higher ln(Aβ$_{42/40}$) was associated with better verbal fluency (WF: $b = 0.18$; 95% CI = [0.01; 0.34]; $p = 0.041$). Higher ln(pTau-181) was associated with poorer learning (B-SEVLT-Sum: $b = -0.09$; 95% CI = [−0.16; −0.01]; $p = 0.021$) and memory (B-SEVLT-Recall: $b = -0.09$; 95% CI = [−0.17; −0.01]; $p = 0.021$). Higher ln(NfL) was associated with poorer learning ($b = -0.08$; 95% CI = [−0.15; −0.01]; $p = 0.029$) and executive functioning/processing speed (DSS: $b = -0.11$; 95% CI = [−0.17; −0.05]; $p < 0.001$). ln(GFAP) was not statistically associated with domain-specific cognitive performance (Fig. 1; Supplementary Data 2).

### Age group stratification

Among adults <60 years, higher ln(NfL) was consistently associated with lower scores on global cognition (GC: $b = -0.09$; 95% CI = [−0.16; −0.03]; $p = 0.003$), learning (B-SEVLT-Sum: $b = -0.09$; 95% CI = [−0.18; −0.00]; $p = 0.049$), and executive functioning/processing speed (DSS: $b = -0.17$; 95% CI = [−0.25; −0.09]; $p < 0.001$) (Fig. 2; Supplementary Data 3).

In adults aged 60–69 years, higher ln(Aβ$_{42/40}$), was positively associated with global cognition (GC: $b = 0.17$; 95% CI = [0.01; 0.34]; $p = 0.034$), verbal fluency (WF: $b = 0.26$; 95% CI = [0.04; 0.48]; $p = 0.023$) and executive

**Table 1 | Descriptive statistics of the Study of Latinos-Investigation of Neurocognitive Aging (SOL-INCA) target population overall and by age group**

| | Overall (N = 5730) | <60 Years (N = 2416) | 60–69 Years (N = 2346) | 70+ Years (N = 968) | P value |
|---|---|---|---|---|---|
| Age | 63.5 (8.2) | 55.2 (2.5) | 64.2 (3.1) | 74.6 (2.7) | $<1.00 \times 10^{-307}$ |
| **Sex** | | | | | |
| Female | 3669 (53.8) | 1516 (51.5) | 1537 (54.1) | 616 (56.7) | 0.099 |
| Male | 2061 (46.2) | 900 (48.5) | 809 (45.9) | 352 (43.3) | |
| **Education** | | | | | |
| Less than high school | 2365 (38.3) | 827 (31.4) | 1013 (39.6) | 525 (46.6) | $3.23 \times 10^{-07}$ |
| High school or equivalent | 1179 (20.4) | 555 (23.6) | 477 (20.0) | 147 (16.2) | |
| More than high school | 2186 (41.3) | 1034 (44.9) | 856 (40.3) | 296 (37.2) | |
| **Hispanic/Latino background** | | | | | |
| Dominican | 506 (9.3) | 227 (10.2) | 197 (9.2) | 82 (8.0) | $5.63 \times 10^{-07}$ |
| Central American | 570 (7.3) | 259 (8.1) | 238 (7.6) | 73 (5.7) | |
| Cuban | 990 (26.9) | 391 (22.9) | 412 (24.6) | 187 (35.8) | |
| Mexican | 2220 (32.8) | 985 (37.1) | 893 (34.6) | 342 (23.8) | |
| Puerto Rican | 931 (15.0) | 337 (12.8) | 394 (15.1) | 200 (17.9) | |
| South American | 411 (5.2) | 171 (5.1) | 171 (5.2) | 69 (5.4) | |
| More than one/other | 102 (3.7) | 46 (3.7) | 41 (3.7) | 15 (3.4) | |
| **BMI** | | | | | |
| Underweight | 26 (0.5) | 7 (0.2) | 14 (0.9) | 5 (0.3) | 0.093 |
| Normal | 890 (16.7) | 360 (15.4) | 353 (16.1) | 177 (19.3) | |
| Overweight | 2287 (40.5) | 965 (41.3) | 926 (40.2) | 396 (39.6) | |
| Obese | 2527 (42.4) | 1084 (43.1) | 1053 (42.8) | 390 (40.8) | |
| **Diabetes status** | | | | | |
| Normal glucose regulation | 966 (16.3) | 517 (21.0) | 347 (15.9) | 102 (9.8) | $6.06 \times 10^{-15}$ |
| Impaired glucose tolerance | 2717 (48.0) | 1204 (51.5) | 1097 (46.6) | 416 (44.6) | |
| Diabetes | 2047 (35.8) | 695 (27.5) | 902 (37.5) | 450 (45.6) | |
| Hypertension | 3190 (58.3) | 1000 (41.4) | 1435 (61.6) | 755 (79.0) | $3.36 \times 10^{-38}$ |
| Dyslipidemia | 2017 (36.7) | 901 (39.1) | 827 (37.4) | 289 (32.3) | 0.016 |
| CKD | 896 (17.4) | 245 (10.7) | 372 (15.8) | 279 (29.6) | $8.20 \times 10^{-22}$ |
| **APOE ε4 genotype** | | | | | |
| ε4 non-carriers | 4482 (78.2) | 1894 (78.0) | 1830 (77.3) | 758 (79.8) | 0.468 |
| ≥1 ε4 allele | 1248 (21.8) | 522 (22.0) | 516 (22.7) | 210 (20.2) | |

Note 1: Results are derived using data from SOL-INCA (unweighted n = 5730) using survey-weighted means and proportions, with differences across age groups assessed using adjusted F-tests and chi-square tests (two-sided tests).

Note 2: All reported values are weighted to represent the SOL-INCA target population, except for unweighted sample sizes (N).

Note 3: Values denote mean (SD) or number (%).

Note 4: No multiple comparison adjustment is applied given the descriptive purpose of the table.

*APOE* Apolipoprotein E, *ε* allele, *BMI* body mass index, *CKD* chronic kidney disease, *SD* standard deviation.

functioning/processing speed (DSS: $b = 0.19$; 95% CI = [0.02; 0.37]; $p = 0.032$). Higher ln(pTau-181) and ln(NfL) were associated with poorer global cognition (GC $_{pTau-181}$: $b = -0.14$; 95% CI = [−0.22; −0.05]; $p = 0.001$; GC $_{NfL}$: $b = -0.10$; 95% CI = [−0.16; −0.03]; $p = 0.006$), and executive functioning/processing speed (DSS $_{pTau-181}$: $b = -0.19$; 95% CI = [−0.29; −0.09]; DSS $_{NfL}$: $b = -0.16$; 95% CI = [−0.24; −0.09]; all $p < 0.001$). Higher ln(pTau-181) was also associated with poorer verbal fluency (WF: $b = -0.18$; 95% CI = [−0.30; −0.07]; $p = 0.002$).

Among those 70+ years, higher ln(pTau-181), ln(NfL), and ln(GFAP) were associated with poorer learning (B-SEVLT-Sum $_{pTau-181}$: $b = -0.23$; 95% CI = [−0.39; −0.08]; $p = 0.004$; B-SEVLT-Sum $_{NfL}$: $b = -0.24$; 95% CI = [−0.43; −0.04]; $p = 0.018$; B-SEVLT-Sum $_{GFAP}$: $b = -0.36$; 95% CI = [−0.57; −0.14]; $p = 0.001$). Higher ln(GFAP) was associated with poorer global cognition (GC: $b = -0.21$; 95% CI = [−0.37; −0.05]; $p = 0.012$), and memory (B-SEVLT-Recall: $b = -0.29$; 95% CI = [−0.50; −0.07]; $p = 0.008$).

## Stratified analyses by cognitive status

We report descriptive statistics by cognitive status in Supplementary Data 4. Among cognitively unimpaired individuals, higher ln(NfL) was associated with poorer executive functioning/processing speed (DSS: $b = -0.07$; 95% CI = [−0.13; −0.01]; $p = 0.028$) (Fig. 3; Supplementary Data 5). Among individuals with MCI, higher ln(pTau-181), ln(NfL), and ln(GFAP) were consistently associated with poorer global cognition (GC $_{pTau-181}$: $b = -0.19$; 95% CI = [−0.31; −0.07]; $p = 0.002$; GC $_{NfL}$: $b = -0.13$; 95% CI = [−0.26; −0.01]; $p = 0.035$; GC $_{GFAP}$: $b = -0.22$; 95% CI = [−0.37; −0.07]; $p = 0.005$), and verbal learning (B-SEVLT-Sum $_{pTau-181}$: $b = -0.28$; 95% CI = [−0.47; −0.09]; $p = 0.004$; B-SEVLT-Sum $_{NfL}$: $b = -0.26$; 95% CI = [−0.44; −0.07]; $p = 0.006$; B-SEVLT-Sum $_{GFAP}$: $b = -0.35$; 95% CI = [−0.56; −0.14]; $p = 0.001$) (Fig. 3; Supplementary Data 5). Higher ln(pTau-181) was associated with lower verbal fluency (WF: $b = -0.21$; 95% CI = [−0.38; −0.04]; $p = 0.016$), and higher ln(GFAP) was associated with poorer memory (B-SEVLT-Recall: $b = -0.21$; 95% CI = [−0.42; −0.01]; $p = 0.044$) and

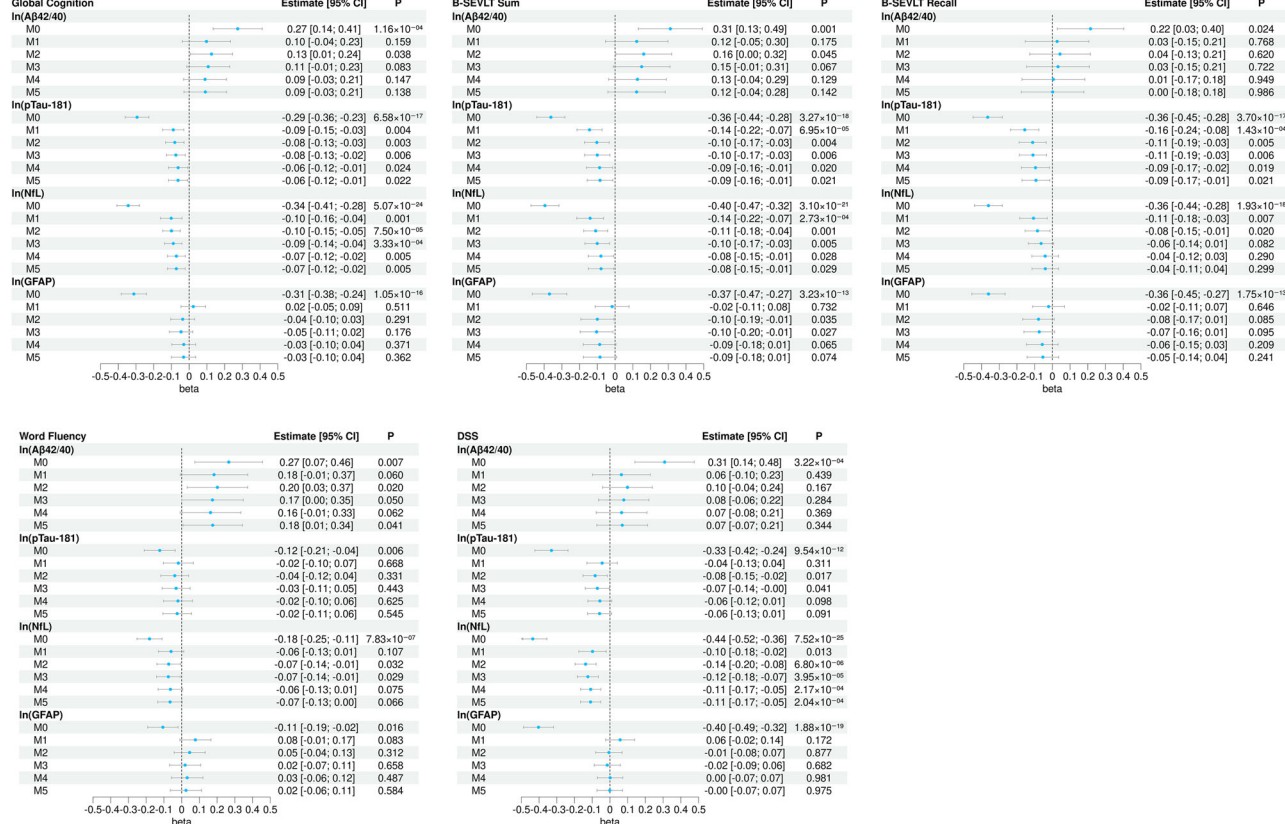

**Fig. 1 | Associations between log-transformed plasma biomarkers and global and domain-specific cognitive performance in the SOL-INCA target population.** Note 1: Results are derived from survey-weighted linear regression models (two-sided tests) using data from SOL-INCA (unweighted $n = 5730$). Note 2: Global cognition score is average of z-scored domain-specific cognitive test scores (M[SD] = −0.00[0.79]). Note 3: M0 is an unadjusted model; M1 is adjusted for age; M2 is additionally adjusted for sex, education, Hispanic/Latino background, and field center; M3 is additionally adjusted for BMI, diabetes, hypertension, and dyslipidemia; M4 is additionally adjusted for CKD; and M5 is additionally adjusted for APOE ε4 genotype. Note 4: Points depict survey-weighted adjusted marginal means and error bars depict 95% confidence intervals. Note 5: Values near zero may appear as ±0.00. Note 6: False discovery rate (Benjamini-Hochberg) was applied as a sensitivity analysis within each outcome. Abbreviations: Aβ amyloid-beta, b regression coefficient, B-SEVLT Brief-Spanish English Verbal Learning Test, CI confidence interval, DSS Digit Symbol Substitution, GFAP glial fibrillary acidic protein, NfL neurofilament light, pTau phosphorylated tau.

executive functioning/processing speed (DSS: $b = -0.24$; 95% CI = [−0.41; −0.06]; $p = 0.007$).

### Associations when all biomarkers modeled simultaneously
In the models where all plasma biomarkers were modeled simultaneously, only NfL remained significantly associated with cognitive performance (Supplementary Data 6; Supplementary Figs. 6, 7). Higher ln(NfL) was associated with poorer global cognition (GC: $b = -0.06$; 95% CI = [−0.13; −0.00]; $p = 0.04$), verbal fluency (WF: $b = -0.09$; 95% CI = [−0.18; −0.01]; $p = 0.026$), and executive functioning/processing speed (DSS: $b = -0.12$; 95% CI = [−0.20; −0.05]; $p = 0.002$).

### Sensitivity analyses
Sensitivity analyses using raw (non-log-transformed) plasma biomarker values and excluding participants with extreme values (>3 SD from the mean) yielded results consistent with primary models (Supplementary Data 7). Findings were also similar in analyses restricted to participants with NfL concentrations measured via the multiplex panel ($n = 5562$), supporting the robustness of the associations (Supplementary Table 3).

### Discussion
In this large, multi-center, community-based sample of middle-aged and older Hispanic/Latino adults living in the US, higher plasma concentrations of plasma biomarkers associated with AD pathophysiology and broader neurodegenerative processes were significantly associated with poorer global cognitive performance. The associations remained after adjusting for sociodemographic characteristics, cardiometabolic health, CKD status, and APOE ε4 genotype and extended to specific cognitive domains such as verbal episodic learning, memory, and processing speed/executive functioning. Although amyloid-beta (Aβ$_{42/40}$) and GFAP were not consistently associated with cognitive performance overall, stratified analyses by age and cognitive status revealed that associations were largely age and cognitive status dependent. These findings support the utility of BBMs relevant to AD and ADRDs in heterogenous populations and emphasize the importance of age and cognitive status.

Our findings are broadly consistent with previous studies in predominantly non-Hispanic White populations, such as those from the Alzheimer's Disease Neuroimaging Initiative (ADNI) and the Mayo Clinic Study of Aging, which have demonstrated that elevated levels of plasma pTau-181 and NfL are associated with worse cognitive performance and greater risk of ADRDs[12,46]. Prior research in Hispanic/Latino populations has largely relied on smaller, region-specific samples such as Caribbean Hispanics in WHICAP or Mexican Americans in the HABLE and HABS-HD studies[20,22,47–49]. For instance, HABLE reported associations between higher NfL and poorer cognitive performance across multiple domains, including processing speed, attention, executive function, and memory when covarying for age, sex, and education[20,47]. Our study expands this literature by examining these associations in a large, multi-center, community-based cohort of middle-aged and older Hispanic/Latino adults from four US metropolitan areas capturing varied backgrounds.

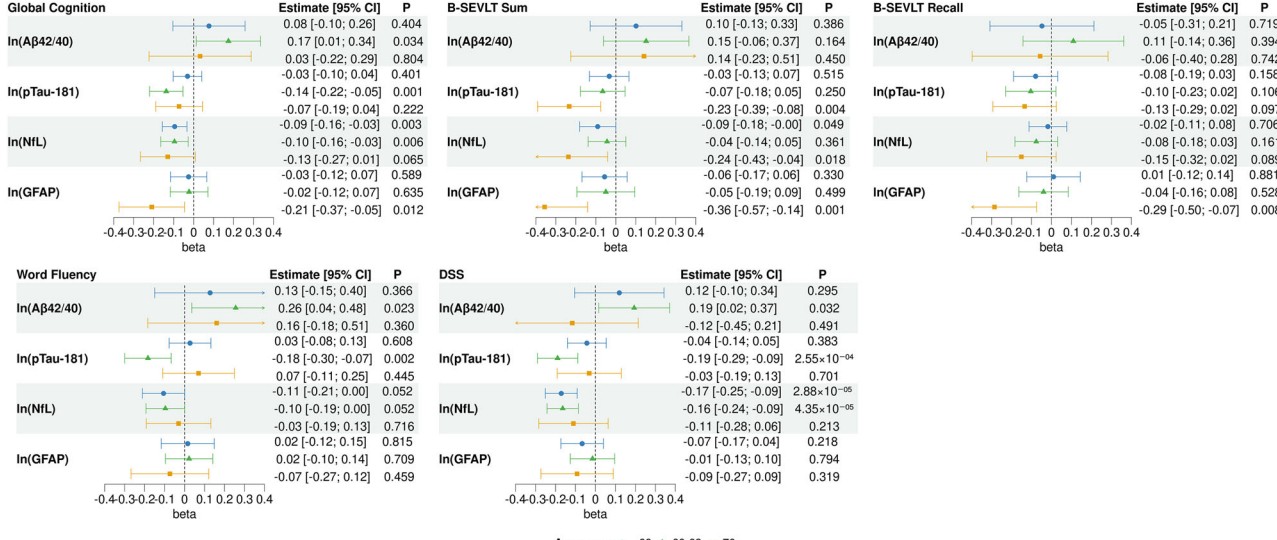

**Fig. 2 | Associations between log-transformed plasma biomarkers and cognitive performance in the SOL-INCA target population, stratified by age group.** Note 1: Results are derived from survey-weighted linear regression models (two-sided tests) using data from SOL-INCA (unweighted *n* = 5730). Note 2: Global cognition score is average of z-scored domain-specific cognitive test scores (M[SD] = −0.00[0.79]). Note 3: Results are from the full covariates models adjusted for sex, education, Hispanic/Latino background, field center, BMI, diabetes, hypertension, dyslipidemia, CKD, and APOE ε4 genotype. Note 4: Points depict survey-weighted adjusted marginal means and error bars depict 95% confidence intervals. Note 5: Values near zero may appear as ±0.00. Note 6: No multiple comparison adjustment is applied. Abbreviations: Aβ amyloid-beta, b regression coefficient, B-SEVLT Brief-Spanish English Verbal Learning Test, CI confidence interval, DSS Digit Symbol Substitution, GFAP glial fibrillary acidic protein, NfL neurofilament light, pTau phosphorylated tau.

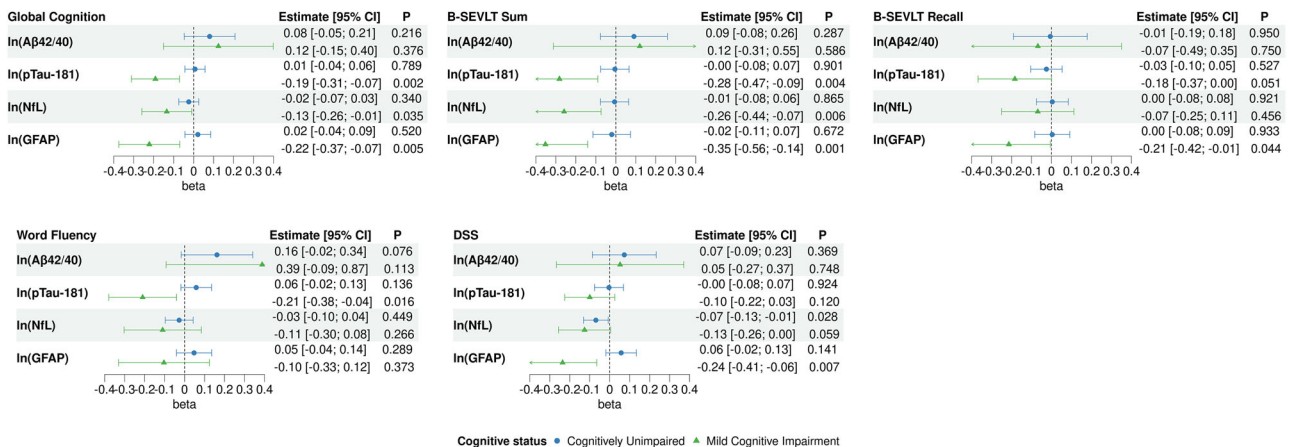

**Fig. 3 | Associations between log-transformed plasma biomarkers and cognitive performance in the SOL-INCA target population, stratified by cognitive status.** Note 1: Results are derived from survey-weighted linear regression models (two-sided tests) using data from SOL-INCA (unweighted *n* = 5730). Note 2: Global cognition score is average of z-scored domain-specific cognitive test scores (M[SD] = −0.00[0.79]). Note 3: Results are from the full covariates models adjusted for age, sex, education, Hispanic/Latino background, field center, BMI, diabetes, hypertension, dyslipidemia, CKD, and APOE ε4 genotype. Note 4: Points depict survey-weighted adjusted marginal means and error bars depict 95% confidence intervals. Note 5: Values near zero may appear as ±0.00. Note 6: No multiple comparison adjustment is applied. Abbreviations: Aβ amyloid-beta, b regression coefficient, B-SEVLT Brief-Spanish English Verbal Learning Test, CI confidence interval, DSS Digit Symbol Substitution, GFAP glial fibrillary acidic protein, NfL neurofilament light, pTau phosphorylated tau.

Importantly, we extended prior research by reporting associations between plasma pTau-181 and GFAP and cognitive performance, which were not previously characterized in this population.

Age emerged as a dominant factor in the associations between plasma biomarker and cognitive performance. This finding aligns with previous research, which suggests that these biomarkers are correlated with age[36,37]. In our analysis, the associations were most attenuated after adjusting for age, with minimal additional changes after adjusting for sociodemographic characteristics and health factor covariates. This suggests that age plays a key role between plasma biomarkers and cognition and the associations are robust to further covariate adjustment. Although BMI, cardiometabolic

health, and CKD were included as covariates in our models, we did not investigate their direct associations with biomarker concentrations or cognitive performance. Prior studies, including our own, have reported that CKD is associated with higher levels of plasma pTau-181, NfL, and GFAP, and lower Aβ42/40, poorer cognitive function, accelerated cognitive decline over 7 years, and higher MCI prevalence[38,40]. Future studies should examine other factors more explicitly, to better understand how cardiometabolic dysfunction may influence biomarker expression and cognition, particularly through subgroup or interaction models.

We did not include a non-Hispanic White comparator group. However, prior studies suggest plasma biomarker levels may vary by ancestry. In

the Mayo Clinic Jacksonville Alzheimer's Disease Research Center, which includes African American and non-Hispanic White participants, CKD and vascular risk factors, but not self-reported race, contributed to variation in plasma AD biomarkers[11]. These findings underscore the importance of considering ancestry, health factors, and social determinants of health in interpreting biomarker levels. Within Hispanic/Latino populations, APOE-ε4 allele varies by genetic ancestry, with a lower prevalence among those with greater Amerindian ancestry, which may modify associations with cognitive performance[50–52]. Although our sample size is large and heterogeneous, we did not stratify analyses by ancestry due to concerns about statistical power and model complexity. Future studies with larger subgroup representation are needed to investigate potential ancestry-related differences in plasma biomarker associations with cognitive performance.

Stratified analyses by age and cognitive status provided important context for interpreting the associations. Among younger (<60 years) and cognitively unimpaired individuals, NfL was consistently associated with cognitive performance, particularly with executive function/processing speed, domains supported by white matter integrity[53,54]. This supports that NfL may serve as an early marker of neuroaxonal damage before clinical impairment becomes apparent. In contrast, older adults (ages 60–69 and 70+) and those with MCI showed broader associations involving pTau-181 and GFAP, in addition to NfL. In those 70+ and those with MCI, GFAP in particular, was consistently associated with global cognition, and episodic verbal learning and memory. These findings suggest that the clinical relevance of these biomarkers may be greater in later stages of cognitive aging or during prodromal disease, and as pathological burden increases, multiple biomarkers, including those related to tau pathology and glial activation, may provide complementary information. Moreover, NfL explained unique variance not explained by other biomarkers, suggesting that it may be a more specific indicator of neurodegeneration. This underscores the potential utility of different biomarkers across disease stages and highlights NfL's promise as an early indicator of neurodegenerative processes in at-risk populations.

Although our findings are consistent with prior literature, the associations were modest, reflecting the multifactorial nature of cognitive function and the cross-sectional design. These results likely reflect early or subclinical relationships rather than diagnostic utility. The weak and inconsistent associations between $A\beta_{42/40}$ and cognitive performance, particularly its lack of association in the fully adjusted models, may indicate that amyloid pathology alone is insufficient to detect early cognitive changes in a heterogeneous, community-based sample. This may be due in part to the temporal lag between amyloid accumulation and clinical symptoms. While the Simoa platform is sensitive, variability across plates, small fold change, and limited dynamic range and sensitivity to detect early amyloid pathology[55]. In contrast, plasma pTau-181 has shown stronger associations with both tau-PET and amyloid-PET in prior studies, and may serve as a more robust indicator of AD pathology across disease stages[56,57].

Some limitations should be noted. First, we lacked CSF and PET imaging and validated thresholds were not established for the biomarkers considered here. Second, longitudinal follow-up is needed to evaluate prognostic and predictive value. Third, although statistically significant, the small regression coefficients may limit immediate clinical translation. Fourth, while the analyses were hypothesis-driven and focused on a limited number of pre-specified biomarkers and cognitive scores no formal adjustments for multiple comparisons were initially applied. We found some attenuation of the associations when accounting for multiple comparisons. We interpret the uncorrected findings as potentially meaningful, but recognize that they should be interpreted cautiously and warrant future validation. Fifth, our global cognitive composite used in this study is conceptually aligned with the Preclinical Alzheimer Cognitive Composite (PACC; a validated summary score used in AD research)[58,59], and consistent with previous HCHS/SOL and SOL-INCA publications, but does not assess of all domains (e.g., attention, visuospatial processing). Sixth, self-identified Hispanic/Latino identity encompasses heterogeneous and admixed ancestry, introducing variability. Lastly, the participants reflect typical community health profiles and were not screened for psychiatric or neurologic disorders, which may introduce additional variability. Despite these limitations, this is the largest study to date of BBMs and cognitive performance in a heterogeneous and varied sample of Hispanics/Latinos in the US. Inclusion of participants aged 50 and over allows investigation of early markers and age-related changes before clinical impairment. Our findings contribute to a growing body of evidence supporting the utility of BBMs in ADRD research and underscore the importance of including heterogeneous populations to ensure generalizability of future diagnostic and therapeutic strategies.

This study provides evidence that plasma biomarkers associated with AD pathophysiology, including $A\beta_{42/40}$ and pTau-181, and biomarkers of broader neurodegenerative processes (NfL and GFAP) are associated with cognitive performance among middle-aged and older Hispanic/Latino adults. These associations are evident across specific cognitive domains and varied by age and cognitive status, highlighting the potential value of plasma biomarkers for identifying early cognitive vulnerability and monitoring neurodegeneration in heterogeneous populations. Our findings underscore the importance of including underrepresented groups in biomarker research for AD detection and intervention. Future longitudinal studies are needed to clarify the predictive utility of these biomarkers and to explore how cardiometabolic health, kidney function, and genetic ancestry may shape their expression and clinical relevance in this population.

## Data availability
The source data for Figs. 1–3 can be found in Supplementary Data 2, 3, and 5. The data supporting the findings of this study are available on request from the HCHS/SOL website: https://sites.cscc.unc.edu/hchs.

## Code availability
The Stata and R scripts used for data cleaning, analysis, and visualization are available to share through Gitlab upon request, with cohort guidelines and permissions for data access.

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

## Acknowledgements
The authors thank the Field Center staff and participants of HCHS/SOL and SOL-INCA staff for their important contributions. This work is support by National Institute on Aging (R01AG075758). H.M.G. also receives additional support from P30AG062429. A.M.S. receives additional support from K08 AG075351 and L30 AG074401. The Hispanic Community Health Study/Study of Latinos is a collaborative study supported by contracts from the National Heart, Lung, and Blood Institute (NHLBI) to the University of North Carolina (HHSN268201300001I/N01-HC-65233), University of Miami (HHSN268201300004I/N01-HC-65234), Albert Einstein College of Medicine (HHSN268201300002I/N01-HC-65235), University of Illinois at Chicago (HHSN268201300003I/N01- HC-65236 Northwestern Univ), and San Diego State University (HHSN268201300005I/N01-HC-65237). The following Institutes/Centers/Offices have contributed to the HCHS/SOL through a transfer of funds to the NHLBI: National Institute on Minority Health and Health Disparities, National Institute on Deafness and Other Communication Disorders, National Institute of Dental and Craniofacial Research, National Institute of Diabetes and Digestive and Kidney Diseases, National Institute of Neurological Disorders and Stroke, NIH Institution-Office of Dietary Supplements. This work was supported by the National Institute on Aging and National Heart Lung Blood Institute. Its contents are solely the responsibility of the authors and do not necessarily represent the official views of the NIH.

## Author contributions
F.M., W.T., and H.M.G. conceptualized the study. F.M., W.T., and S.K. led study design and statistical analyses. H.Z. and J.C. advised on statistical methods and supported model implementation. B.T. and W.T. contributed to data processing and management. W.T., S.K., D.F.V., R.B.L., R.K., and H.M.G. contributed to manuscript drafting and interpretation of findings. F.D.T., A.M.S., N.Z.A., C.R.I., G.A.T., M.D., J.C., H.Z., D.G., L.C.G., C.D., and B.T. provided critical feedback and domain expertise. G.A.T., M.D., L.C.G., C.D., and H.M.G. obtained study funding. W.T., D.G., B.T., L.C.G., C.D., and H.M.G. provided administrative and technical oversight. W.T., D.G., L.C.G., C.D., B.T., and H.M.G. supervised the project. F.M. wrote the paper. All authors critically revised the manuscript for intellectual content and approved the final version.

## Competing interests
The authors declare no competing interests.

## Additional information

**Freddie Márquez** [1] ✉, **Wassim Tarraf**[2], **Sayaka Kuwayama**[1], **Deisha F. Valencia**[1], **Ariana M. Stickel**[3], **Richard B. Lipton** [4], **Fernando D. Testai** [5], **Natasha Z. Anita** [1], **Kevin A. Gonzalez**[1], **Robert Kaplan**[6], **Carmen R. Isasi** [6], **Gregory A. Talavera**[3], **Martha Daviglus** [7], **Jianwen Cai** [8], **Haibo Zhou**[8], **Douglas Galasko** [1,9], **Linda C. Gallo** [3], **Charles DeCarli** [10], **Bharat Thyagarajan** [11] & **Hector M. González**[1]

[1]Department of Neurosciences, University of California San Diego, La Jolla, CA, USA. [2]Institute of Gerontology & Department of Healthcare Sciences, Wayne State University, Detroit, MI, USA. [3]Department of Psychology, San Diego State University, San Diego, CA, USA. [4]Department of Neurology, Albert Einstein College of Medicine, Bronx, NY, USA. [5]Department of Neurology and Rehabilitation, University of Illinois at Chicago, College of Medicine,

Chicago, IL, USA. [6]Department of Epidemiology & Population Health, Albert Einstein College of Medicine, Bronx, NY, USA. [7]Institute for Minority Health Research, University of Illinois at Chicago, College of Medicine, Chicago, IL, USA. [8]Department of Biostatistics, University of North Carolina at Chapel Hill, Chapel Hill, NC, USA. [9]Shiley-Marcos Alzheimer's Disease Research Center, UC San Diego, La Jolla, CA, USA. [10]Department of Neurology, University of California, Davis, Sacramento, CA, USA. [11]Department of Laboratory Medicine and Pathology, University of Minnesota, Minneapolis, MN, USA. ✉e-mail: frmarquez@ucsd.edu

