## [Transparent Peer Review file · Communications Medicine]

Alzheimer's disease plasma biomarkers are associated with cognitive performance among Hispanic/Latino adults

Corresponding Author: Dr Freddie Márquez

Version 0:

Reviewer comments:

Reviewer #1

(Remarks to the Author)

This study is dealing with blood based biomarkers for Alzheimer's disease, which is very timely topic. This study with a large sample size was well-designed, and the results are succinctly described. Study limitations are also well mentioned. However, the following minor points are suggested to be included in the revised manuscript.

Abstract

- It would be better describe the final sample size.
- Please clarify whether this study is cross-sectional or longitudinal.

Background

- The term "Alzheimer's disease" should be expressed in abbreviated form after the abbreviation.

Method

Data

- As mentioned in the Abstract section, it would be helpful for readers to clarify whether this study is cross-sectional or longitudinal.
- The authors did not mention whether individuals with any psychiatric or neurologic disorders that can affect the cognitive function were included.
- Clinical diagnosis of cognitive status (normal, MCI, or dementia) is also required ,if available.

Blood collection and processing procedures

- "All coefficients of variability of the blinded duplicates were <11.3."  If "coefficients of variability" means "coefficients of variation," please correct the term with the appropriate unit for "11.3".

Covariates

- The term "high school" is expressed as "HS" in Table 1, but not in the manuscript.
- The terms "CKD" and "BMI" are used in abbreviated form in the manuscript, but in full-term in the Table 2. Unifying the expression is recommended.

Analytic Approach

- "First, we characterized the SOL-INCA target population using the covariates included above (Table 1)." -> This sentence seems unnecessary in view of the redundant description at the beginning of the "Results" section.
- "Extreme values (outliers at extreme values above and below the mean) of the plasma biomarkers were excluded..." -> Please clarify the criteria defining the extreme values.

-

Results

- The title "Primary analysis" would be better replaced with a more appropriate title that describes the content of the analyses.
- Table 2 is unnecessarily duplicated in page 21 and page 23.
- Page 9: The abbreviated term "CI" has been used as full term (confidence interval) in the Method section and last part of page 9.

-

Discussion

- Please discuss more thoroughly the issue about lower effect size of Abeta42/40. It could be result from the cross-sectional nature of this study, based on the amyloid cascade hypothesis, or the assay's issue (Simoa technique).
- Individuals were from US metropolitan, which cannot represent the whole Hispanic/Latino group.

Reviewer #2

(Remarks to the Author)

In this study, Márquez et al. investigate the associations between plasma biomarkers for AD and cognitive performance in a large population (N = 5730) of Hispanic/Latino individuals. They show that in this sub-population, plasma biomarkers of AD are associated with various aspects of cognitive performance, but that these were differentially affected when controlling for several demographic factors and comorbidities. The study is important as data regarding plasma biomarkers for AD in Hispanic populations is lacking, especially in such a large sample. Moreover, the manuscript is really well written. However, there are some statistical concerns and additional analyses to be addressed.

Major comments

Methods

1. Line 123: how were the z-scores derived for the global cognitive composite score? Were they based on cognitively unimpaired individuals? Please add a brief description.
2. Line 124: "cognitive tests used at Visit 2", does that mean that different tests were used for the visit 2 data? Were visit 1 and visit 2 z-scores used interchangeably? Please clarify in the manuscript to avoid confusion.
3. The statistics should be described more in depth. Several things are missing:
 - a. Line 175 mentions that outliers were defined as extreme values, but how were these extreme values defined?
 - b. Did you address the regression assumptions for the linear regression models? I.e., were biomarker levels log-transformed, was normality checked, etc.? Please add to the manuscript. Related to this: in Figure 1 it looks like plasma biomarker levels were also z-scored, but this is nowhere explained. Please explain how z-scoring was done for the biomarkers.
4. A lot of models are being executed here but nothing is mentioned about correcting for multiple comparisons. Hence, I would suggest correcting for multiple comparisons here per cognitive test per model (so within your Global Cognition models, correct for the 4 biomarker models you conduct within M1, and then for the 4 within M2, etc.), which will probably mean that some of the weaker p-values will become insignificant.

Results

1. Please add correlation plots between cognitive performance and raw plasma biomarker levels (e.g. to supplement, or as main figure 1).
2. Please add a M0: show the effects of just age, sex, education, and Hispanic/Latino background on cognition. Perhaps these already contribute quite a lot to differences in cognition?
3. You could consider displaying the models in Figure 1 in a forest plot rather than as linear models and show the beta coefficient of the biomarker for each model. That would show nicely how the beta changes between models.
4. Please report R² for the models in addition to std. B (95% CI). To see whether one model is significantly better than the other, I'd suggest comparing the R²'s between models with a bootstrap-hypothesis test for instance, or to compare models based on their AIC.
5. Please stratify analyses between CU and CI individuals, if you have cognitive status data available (unclear whether that is the case). This would be valuable information. Please also add cognitive status as a variable to Table 1, and add a supplementary table split between cognitive groups, if sample sizes allow it.
6. With your large sample size, it would be interesting to see what happens when adding all biomarkers to the same model with cognition as outcome and see if they all remain significantly associated. You should have sufficient power to try this out! Can be a supplementary figure/table.

Minor comments

1. Please mention the sample size in the abstract.
2. Table 1: I would suggest formatting the demographics for categorical variables as n (%), this is easier to interpret and highlights your large sample size (which is a strength!).
3. Line 61: these non-specific biomarkers are involved in both AD and other brain diseases, please correct.
4. Line 76: please clarify in what way the APOE-e4 distribution varies among Hispanics/Latinos, and why that is relevant. Also in line 277: why is the effect of APOE-e4 especially relevant here?
5. Please add a section to the discussion regarding the small beta's for the plasma biomarkers. The effects of the plasma biomarkers on cognition seem to be rather small.
6. Please add to the limitations that your sample consists of self-identified Latino's which can introduce some bias

Reviewer #3

(Remarks to the Author)

Thank you for the opportunity to review this manuscript by Márquez et al., which describes the association between AD plasma biomarkers and cognitive performance in a large cohort of US Latinos. Blood biomarkers have emerged as accessible and highly promising tools for advancing the diagnostics of AD. However, more data on how these biomarkers may vary across diverse ancestries is critical in order to ensure broad and equitable translation to clinical practice. The manuscript has some weakness in the presentation and discussion of the results. Specifically, I have the following concerns that need to be addressed.

Background.

Lines 65-66: the sentence is not supported by a pertinent citation. In addition, I suggest to briefly describe the literature data on the validity of blood biomarkers in clinical practice.

Lines 79-81: I suggest to add a brief description of the literature data on blood biomarkers in Hispanic/Latino populations, clarifying the gap that the present study would fill in.

Lines 85-87: The pre-specified hypothesis is in line with what know on blood biomarkers in non-Hispanic white people. Did the Authors have hypotheses reflecting the peculiarity of the population under study?

Minor:

Line 48: 'deposition' should be more appropriate than 'formation'.

Line 56: 'Associated' should be more appropriate than 'linked'.

Line 69: I would suggest to use the term 'Relevant/important' instead of 'a few'.

Methods.

The information about the proportion of MCI is of great relevance and should be reported. According to González et al. *Alzheimers Dement* 2019, the data is available. The regression models should be run separately in cognitively intact and MCI populations to overcome the limitation recognized in the Discussion.

Pag. 5, lines 107-109: for clarity, I suggest to move the information on excluded data in the pertinent paragraphs, i.e., blood collection (n=151, samples not collected), covariates (n=496) and statistical analysis (outliers). Please, specify what is meant by 'missing exposure' in supplemental figure 1.

Pag. 6, line 123: The Authors refer to a description of the z-score computation, which is not presented.

Lines 161-162: the value of triglycerides used to determine the presence of dyslipidemia is not specified.

Lines 170-171: the reference to the table 1 is not appropriate. Herein the Authors should specify the statistics used to describe the population.

Line 173: the term 'survey' to define a regression model seems unusual. Please specify.

Line 175: the analysis of outliers should be described in more detail (more than n=? standard deviations above or below the mean).

The statistical software used to make the analyses should be reported.

Minor:

Line 168 should be eliminated.

Results.

The raw data of neuropsychological test scores and blood biomarker values should be reported as supplementary tables, for completeness of the data.

In addition, it would be very interesting to show the mean levels of each of the biomarkers compared to non-Hispanic white individuals; indeed, the influence of ancestries on AD blood biomarkers is not fully understood (see doi: 10.1212/WNL.000000000207675).

The table 2 should report number(%). Age should be reported at the top of the table together with other sociodemographic features. In order to lighten the table, mean BMI and non-APOE4 allele frequencies can be moved in the text.

Lines 192-216: the paragraph is very difficult to read. The Authors should describe the results for each biomarker in separate paragraphs. In addition, the information in the text should not duplicate that reported in the table 2. In the text, the Authors should highlight the most significant results (e.g., the associations that remain significant in the full adjusted model).

The figure 2 is hard to read: the models from M2 to M5 are overlapped.

Discussion.

Lines 228-230: the term 'attenuated' seems incorrect: the associations are not significant in adjusted models. The Authors should summarize the results in a more coherent and concise manner, e.g., first reporting significant associations with global cognition, and then detailing which cognitive domain the association is driven by, for each blood biomarker.

Lines 249-254: these considerations have been already reported in the Introduction. They can be omitted.

Line 267: the term 'cognitive outcome' is appropriate for clinical trials, not population-based studies.

Lines 269-270: Has a measure of cognitive reserve been collected? What 'environmental influences' refer to? Please clarify.

Lines 270-275: the hypothesis that the association between lower plasma Aβ_{42/40} and reduced verbal fluency can be underlined by CCA is interesting but quite speculative. This should be made clearer. To my best knowledge, verbal fluency do not rely on processing speed. A reference for this statement is not provided.

Lines 287-288: this sentence seems to contradict the following one: neuroinflammation can disrupt circuits essential for executive functioning, but GFAP, a marker of inflammation, is associated with memory, not executive functions.

Lines 290-291: this sentence is quite general and it is not supported by a citation.

The implication of the results for clinical practice should be addressed.

Reviewer #4

(Remarks to the Author)

The manuscript titled "Plasma biomarkers of Alzheimer's disease and related dementias and cognitive performance among Hispanics/Latinos: Findings from the HCHS/SOL and SOL-INCA" explores the relation between neurodegeneration and Alzheimer's disease plasma biomarkers with cognition scores in a cohort of Hispanic and Latino people living in multiple US cities. The measurements are taken on the ultrasensitive SIMOA platform and on an impressive amount of patients, >5700, and they are also well characterized with extensive information on each patient of both a socioeconomic and health nature. The aim of the study is clearly stated in the abstract and introduction and the authors do a good job in not digressing from the aim and fulfilling it, as they show some associations between the plasma biomarker levels and the cognitive scores. The manuscript is well written and the introduction and methods are clear to follow. Overall, the article could contribute to the field by reporting measurements on a population that has been previously under-represented in these biomarker studies.

Major comments:

1. The manuscript's strength of a very high number of participants, comes with statistical challenges that were not addressed. All 5700+ patients were treated as one group, this is somewhat unusual, since often large cohorts are segmented in subgroups, either diagnostic or demographic ex. age brackets. This approach would also reduce the chances of having significant P-values that are not clinically meaningful. The reported Beta coefficients and confidence intervals in Table 2 are statistically significant but very close to 0. Have the authors considered dividing the population in groups or to validate the associations in a subset of patients?
2. No raw data is shown in the manuscript. The only figure shows estimates from the model. Some correlations between marker and score or violin plots comparing some groups (ex. High-scoring Vs. low-scoring) would be beneficial to further show if there is a relation between the cognitive scores and plasma biomarker levels and their distribution.
3. The rationale for the choices of the covariates for the model is not argued clearly. In Figure 1, it is possible to see how models 2-5 are basically identical in performance. I would suspect that this is driven by adjusting for age, which drives most of the effect, especially considering the wide-spread age span (50-86) present in the samples. It is known how biomarkers, for example NfL (PMID: 35865350) and other biomarkers used in the study (PMID: 37237388), are positively correlated with age. Can the authors clarify if they have used other combinations of models and if a model just adjusted by age would perform the same as M2-M5?
4. A discussion between clinical and statistical significance is lacking in the limitations section of the manuscript. With such small Beta coefficients and such a large sample size it is likely the clinical significance is limited. Unless, the results can be replicated in a smaller subset of samples.
5. For such a large study run on the SIMOA platform (an estimation of >70 plates per assay), one can expect a technical variation of at least around 10%. The manuscript should include whether any control samples were used in the plates to keep track of such variation. If they have been used the values for the repeatability and precision should be included in the manuscript and whether the data has been normalized based on the controls to reduce this variability. Or have any other statistical strategies been used to address this? If they have not been included, this should be acknowledged as a limitation of the study.

Minor comments:

1. Line 78-79: The summarization of REF.12 is not entirely faithful, as the manuscript does not argue that comparisons are not applicable between different socioeconomic backgrounds, rather that it might affect the risk-factors to develop diseases, such as dementia.
2. Line 79-80: But has any similar study been carried out in cohorts representing other ethnicities, what were the findings? Will they differ between this Hispanic and Latino cohort?
3. Line 92: The general description of the cohort says it is composed of individuals ages 18-74, in the abstract however it says the study included participants aged between 50-86. Were there any participants older than 74 then?
4. Line 108- 109: The sentence could benefit from a clarification that also the number of samples measured for each assay is displayed in the Supp. Fig 1., as this information is lacking in the text.
5. Line 111: In the "Neuropsychological assessment" paragraph, it could be helpful to include if there is a range between which patients are classified as having cognitive deficiencies, this would give more context later on to the estimates from the model, if for example a change of 1 in the score is meaningful or one of 10 is.
6. Line 174-177: The specific rationale of the definition of outlier is not clearly identified, other than a vague statement of "(outliers at extreme values above and below the mean)". The statistical rationale should be stated ex. $\pm 3SD$ from the mean. Have the authors considered whether the upper outliers could be individuals with Alzheimer's disease? Moreover, based on the numbers reported of outliers in Supp. Fig 1, their effect against 5700+ "non-outliers" is potentially negligible, has the analysis been conducted with and without outliers?
7. Line 177: The authors should state in the text what the "outcome" is and also clearly state what the predictor variables are, this would also improve the clarity of Figure 1.

8. Line 181- 183: In Figure 1 it is not clear where the implementation of the post-hoc ANOVA is, moreover was the post-hoc ANOVA done to compare the models? Was there any significant change between them? Where is this ANOVA outcome reported? Even if the result is not significant it should be stated in the results.
9. Line 191: The section titled “Primary analysis” would benefit from referring to the models by their name ex. M1, M2 etc. Otherwise, for example in line 196, it is not immediately clear what the “sociodemographic covariates” are.
10. Line 199: In table 2 the beta coefficients are reported as being standardized, it would help the clarity to state it also in the main text and label the β accordingly.
11. Line 251-254: The makeup of the Hispanic/Latino group is described as being very heterogenous, could the authors elaborate on why they did not check if there are differences between the various Hispanic/Latino ancestries present in the cohort? The information is reported as being available in Table 1.
12. Line 264-265: “it is unclear if amyloid-lowering drugs would benefit all populations at risk.” Please provide a reference for this statement or rephrase.
13. Line 266 & 283: the description of the markers as “non-specific” should be rephrased as “markers of neurodegeneration”.
14. Line 277-278: Here APOE genotype is described as attenuating only slightly the relationship between GFAP and cognitive specific performances. Is this a reference to results from Model 5? In this case APOE is added after having adjusted for many covariates, have the authors checked if just adjusting for APOE status (perhaps along with age and sex) also attenuates the association only mildly? So, to increase clarity, the authors should state if it is M4 Vs. M5 or some other comparison.
15. Line 288-290: It is not clear if the association between learning and GFAP is supported by evidence from this study or from data present in REF. 31. Please clarify this sentence.
16. Line 295-296: Could the authors elaborate on why they speculate that “...it is possible that other non-AD prevention and treatment strategies should be sought for Hispanics/Latinos”. Do they intend “other” as in different from other ethnicities or as in lifestyle-changes that are non-specific to AD, ex. Weight loss. And are these specific to just Hispanics/Latinos or could benefit anyone regardless of ethnicity?
17. Line 304-305: If the information regarding patients being MCI or CU is available for participants it should be included in the demographics table. Moreover, how come this important metric was not used in any analysis? Was the MCI status determined with the cognitive tests presented in this manuscript? If so, it would be beneficial to include the cut-offs in the methods section.
18. Line 306: “...the influence of blood-based biomarkers on cognitive performance...” The sentence should be improved in the wording as the blood-based biomarkers do not influence cognitive performance.
19. Line 308-311: Whilst a large spread of ages could be a very interesting proposal and strength of this cohort, in this manuscript any differences regarding age are not presented. Have the population levels of the biomarkers been investigated at different age brackets? Currently, age seems to be included only as a variable in the models, which is therefore used to cancel the effect of age on the results.
20. Figure 1: In the figure legend, it should be made clear which is the outcome variable and which is the predictor variable. Are we seeing the predicted global cognition scores based on biomarker levels or the other way around?
21. Figure 1: In the figure panels, the lines include some dots, what do the dots on the lines represent? In the analysis has the global cognition score been used as a numerical variable or ordinal categorical? Could the authors please clarify.
22. Table 1: In the demographics table, age is only presented as the mean with SD. It would be beneficial to the reader to have it presented also in age brackets (ex. 50-60, 60-70 etc.) so have an idea of the distribution. Similarly, the amount of patients classified as with normal cognition or mild cognitive impairment should be included. As these two metrics are correlated with the levels of the biomarkers measured.
23. Table 2: In the table please adjust the rounding in the cases where -0.00 is used it should read -0.01. This was also present in the manuscript.

Reviewer #5

(Remarks to the Author)

Dear dr. Marquez,

First, I would like to commend you for conducting research on minority populations, who represent a large and frequently understudied group affected by dementia. Your use of a substantial sample size and high-quality Simoa-based measurements underscores the methodological rigor of your work.

I understand your work to focus on:

Investigating associations between AD-specific (A β 42/40, pTau181) and non-AD specific (NfL, GFAP) plasma biomarkers and cognitive performance in a diverse cohort of community-dwelling Hispanic/Latino adults in the USA.

Although the manuscript's title initially piqued my interest, I must note that the content did not fully align with my expectations. Below are my major and minor concerns:

MAJOR CONCERNS

BACKGROUND

- Additional references would strengthen certain claims. For instance, the sentence: "More recently, the ATN framework has been implemented using more accessible and less expensive blood-based biomarkers ..." cites only one source.
- You make mention of criteria for diagnosing AD, but leave out in the 'while alive' and in A-T-N framework (as diagnosing AD postmortem is still the golden standard, but clearly does not need to make use of blood-based biomarkers).
- In the statement: "(3) biomarkers of common non-AD co-pathologies," it would be helpful to specify which co-pathologies you are referring to.
- You note that "(3) the APOE- ϵ 4 allele distribution varies among Hispanics/Latinos," but do not address this stratification in the discussion.
- You state there is no consensus on whether biomarker comparisons apply across socioeconomic strata. This point distracts slightly from your main findings.

DATA

- The participants section is confusing. It is unclear whether this study includes only the SOL-INCA samples or HCHS/SOL. Consider omitting excessive detail about HCHS/SOL to avoid confusion.
- "A total of n=496 participants were excluded due to missing covariates" should specify which covariates. Also clarify the timing and criteria for "exclusion of extreme values."
- Your four neuropsychological tests might be insufficient to represent global cognitive function.
- In a multicenter study, explain how plasma samples were assayed. Was everything run simultaneously on a single Simoa platform? You mention NfL measurements using two different kits (N4PE vs. NF-light advantage). Were bridging samples included, and how was agreement between assays established? Was plasma pTau181 measured in singlicate or duplicate?
- Judging from Figure 1, only the unadjusted model (m1) differs meaningfully from all the adjusted ones. Adding three further models does not seem to contribute much if incremental differences are minimal and under-discussed.
- You mention these covariates were included based on the literature. Did you confirm their relevance in your own dataset?

DISCUSSION

- Please clarify the statement: "In clinic-based studies, there is evidence that supports the risk-stratification capabilities and utility of blood-based biomarkers that detect abnormal protein accumulation in the brain" How does risk-stratification specifically relate to your findings?
- Overall, the discussion might be more focused if it aligned directly with the research question. Consider minimizing tangential points, such as the reference to cerebral amyloid angiopathy (CAA).

MINOR CONCERNS

ABSTRACT

- Mention explicitly that your participants are community-dwelling Hispanic/Latino adults.
- Indicate which cognitive tests were used.
- Avoid phrases like "worse global cognition" for cross-sectional data; "poorer global cognitive performance" is more precise.
- The text says "Elevated plasma biomarker levels of A β and tau ... are associated with lower cognitive performance," but it is actually lower A β 42/40 that associates with poorer performance.

BACKGROUND

- Use abbreviations consistently (i.e., "Alzheimer's disease" versus "AD," and "blood-based biomarkers").

DATA

- Include protocol numbers for the Institutional Review Boards.

RESULTS

- The interplay of covariates sometimes diminishes or accentuates associations, which can be confusing. Consider clarifying these findings.

DISCUSSION

- "GFAP is a sensitive biomarker for detecting and tracking A β pathology at early stages of AD" is likely an understatement. Multiple recent studies strongly support GFAP's role in AD pathophysiology.

- This sentence is quite unclear as you do not state a direction of the associations: "Previously, the Health and Aging Brain among Latino Elders (HABLE) study also reported that higher levels of NfL were associated with several cognitive abilities, including processing speed, attention, executive function, and memory when covarying for age, sex, and education in a cohort that included Mexican-Americans".
- You note that including individuals who are cognitively unimpaired and those with mild impairment could affect biomarker-cognition relationships. Why not stratify or at least discuss this distinction more explicitly?

FIGURES AND TABLES

- Table 1: Indicate total n for each category.
- Table 2 is duplicated; also, clarify which cognitive domains the tests represent.
- With a large dataset, distributions of plasma biomarker values or composite cognitive scores might be quite insightful.
- Consider adding line numbers for easier review.
- In the supplemental figure's flow chart, clarify any points I noted in the attachment.

SUPPLEMENTAL FIGURE

It is always helpful to have a flowchart, thank you for providing this. However, please consider providing the following:

- Abbreviations do not appear in alphabetical order in accompanying text
- Biospecimen collection = blood? (because it could be a myriad of things)
- Which covariates? And which covariates are missing?
- What method was employed for removal of outliers (extreme values based on what)?
- And also, were the outliers above or below the mean? For example, I can imagine NfL having high outliers whereas GFAP would have low outliers

I wish you success with these revisions.

Reviewer #6

(Remarks to the Author)

This study addresses an important and timely topic by examining plasma biomarkers and cognitive performance among Hispanic/Latino adults—an underrepresented population in Alzheimer's disease research. The large sample size is a notable strength and offers a valuable opportunity to advance our understanding of blood biomarker utility within this group. However, several critical issues limit the current manuscript's contribution. Addressing these concerns would significantly strengthen the scientific rigor and relevance of the study.

1. While the authors highlight the underrepresentation of Hispanic/Latino populations, the manuscript does not sufficiently situate its findings within the broader scientific literature. A deeper comparison would help clarify the novelty of the results and determine whether the observed associations are consistent across populations.
2. The cognitive battery (B-SEVLT, Word Fluency, DSS) covers limited cognitive domains. While these tests offer valuable information, they do not represent "global cognition" as they do not fully capture global cognitive domains. Expanding the cognitive assessment to include additional domains—such as attention and visuospatial skills—would provide a more comprehensive understanding of cognition in this population.
3. While CKD and BMI were included as covariates in the regression models, their direct relationships with biomarker levels were not fully explored. Given the distinct health profiles within this population, it would be informative to investigate how these covariates directly influence biomarker levels and cognitive outcomes. This could include subgroup analyses or interaction effects, which would provide deeper insights into these relationships.
4. The finding that higher AD plasma biomarkers are associated with worse cognitive performance is expected. However, given the inclusion of both cognitively unimpaired and MCI individuals, stratified analyses are essential to provide a more nuanced understanding of these associations. Moreover, since the authors identify factors such as CKD and CVD risk as important covariates, stratifying analyses by these variables—alongside cognitive status—would offer deeper insights into how these health conditions may modify biomarker-cognition relationships.
5. Some sections of the discussion include content that is less relevant to the core findings. Focusing more on potential biological mechanisms—especially regarding why biomarker associations were attenuated after adjusting for CKD and CVD risk factors—would strengthen the discussion. For instance, elaborating on how vascular or metabolic health may influence biomarker-cognition associations could add meaningful context. Furthermore, the discussion should integrate relevant literature on racial/ethnic differences in biomarker profiles and cognition to enhance the impact of the findings.

Version 1:

Reviewer comments:

Reviewer #1

(Remarks to the Author)

Authors have revised all points appropriately.
I appreciate the effort from the authors.

I think this manuscript has enough quality for publication in Communications Medicine, after correcting the following minor point.

Page 6

"Blinded duplicated showed coefficients of variation <11.3%. We included ..., demonstrated coefficients of variation (CVs) below 12% across plates." -> "Blinded duplicated showed coefficients of variation (CV) <11.3%. We included ..., demonstrated CVs below 12% across plates."

Reviewer #2

(Remarks to the Author)

The authors have done a great job addressing the long list of comments, and the manuscript has certainly improved a lot. However, not everything has been correctly addressed.

1. I suggested correcting for multiple comparisons per cognitive test per model, as many models are being executed here. The authors explained how they chose not to formally apply corrections here as biomarkers were selected a priori and highlight cautious interpretation in the limitations. However, I disagree. Even if variables were selected based on strong rationale, a lot of different tests are still being executed which gives a higher chance of a type I error occurring, something that still applies regardless of a priori decisions. Please add a column, or row, containing p-values corrected for multiple comparisons, to your Tables 2-4.

2. My comments to add a M0 has been misinterpreted. I asked to show the effects of just age, sex, education and Hispanic/Latino background on cognition. My apologies if I was not clear enough. That would look like: cognition ~ age + sex + education + background. So no biomarkers included in the model. The results could be reported in a supplementary table for the main analyses only. Considering we know these sort of demographic factors have a substantial impact cognition, seeing their effects would improve our understanding of how much additional value biomarkers hold in addition to these demographic factors.

Reviewer #3

(Remarks to the Author)

Manuscript#: COMMSMED-25-0342A

I thank the Authors for addressing my concerns. I have some additional comments.

Major comment:

1. The interchangeable use of the terms AD and ADRD when referring to the plasma biomarkers is a source of confusion. This issue is apparent throughout the manuscript. In the Background section, it was stated that NfL and GFAP are non-specific biomarkers of AD and other brain diseases (which brain diseases? ADRD?). However, in the Discussion section the Authors referred to NfL and GFAP as biomarkers associated with AD pathophysiology (see lines 371-372 for an example). In the Background section, it would be useful to clarify which biomarkers were indicative of AD pathophysiology and which others were best associated with ADRD. The discussion of the results should be consistent with this understanding.

Other comments:

2. Page 4, line 58: please specify that previous research refers to non-Hispanic White populations and add pertinent references.

3. page 4, line 62: please specify that age and cognitive status were also taken into account.

4. A brief explanation on the criteria used to assign a diagnosis of MCI should be added in the paragraph 'Neuropsychological assessment'. The title of the paragraph could be changed into 'Neuropsychological assessment and cognitive status'.

5. Supplementary Figure 1 should report a brief explanation for the terms 'missing exposure' and 'censored'. The same terms should also be referred to in the text (pag.6, lines 110-113).

6. Pag.8: the sentence 'we interpreted findings cautiously in the context of multiple testing' is unclear. Please specify.

7. The number of MCI should be reported in the paragraph 'Target population characteristics.'

8. Table 1: the second and fourth lines [Mean (SD), N (%)] may be eliminated. Instead, the sentence 'Values denote mean(SD) or number(%)' may be added to the notes. As regards hypertension, dyslipidemia, and CKD status, only the values referred to the presence of the conditions should be reported. APOE4: 'negative' should be 'non carriers'.

9. Table 3 and 4 overlap with Figure 2 and 3. I suggest to move them into the supplemental material.

Reviewer #4

(Remarks to the Author)

To the kind attention of Dr.Marquez et al.,

I would like to congratulate you with the revision work of the manuscript, especially considering you received five reviewer reports and you had addressed all comments very thoroughly. The comments I provided were all addressed with adjusting the text, new statistical analyses and new figures. Therefore I am satisfied with the updated version of the manuscript. The added insight provided by the stratification of patients and added figures have elevated the insights provided by the

manuscript.

Reviewer #5

(Remarks to the Author)

Dear Dr. Marquez,

Thank you for your thorough revisions. I appreciate your diligent efforts in addressing the points raised in my review. I am satisfied with the changes made to the manuscript, particularly the improvements in clarity, methodological detail, and the discussion of key findings.

While a few minor points of discrepancy remain, they do not detract from the overall quality and enhanced rigor of the work. I believe the manuscript has been significantly strengthened and is now well-positioned for publication.

Reviewer #6

(Remarks to the Author)

The revised manuscript provides a much-improved contextualization of the findings, with a clear comparison to prior studies in non-Hispanic White, African-American, and Hispanic/Latino cohorts. This addition effectively highlights both the novelty and consistency of your results. Your justification for the cognitive composite—positioned as analogous to the PACC—and the acknowledgment of its limited domain coverage are appropriate and adequately address my earlier concern. Retaining CKD and BMI as a priori covariates, while reserving interaction analyses for future work, is a reasonable decision given the cross-sectional nature of the study. Most notably, the addition of stratified analyses by age (<60, 60–69, ≥70 years) and cognitive status (CU vs. MCI), with results clearly presented and referenced, fully addresses my request regarding potential effect heterogeneity. I appreciate the thoughtful revisions and have no further comments.

Version 2:

Reviewer comments:

Reviewer #2

(Remarks to the Author)

All comments have been addressed, well done!

Reviewer #3

(Remarks to the Author)

I thank the Authors for their great job in revising the manuscript. I do not have further comments.

Reviewers' comments:

Reviewer #1 (Remarks to the Author):

This study is dealing with blood based biomarkers for Alzheimer's disease, which is very timely topic.

This study with a large sample size was well-designed, and the results are succinctly described.

Study limitations are also well mentioned.

However, the following minor points are suggested to be included in the revised manuscript.

Abstract

- It would be better describe the final sample size.

Thank you for the suggestion. We have revised the Abstract and Methods sections to clearly report the sample size and include a detailed chart in supplement.

- Please clarify whether this study is cross-sectional or longitudinal.

Thank you for pointing this out. We have clarified in both the Abstract and Methods sections that this is a cross-sectional study based on data collected during HCHS SOL Visit 2 (2014-2017) and its ancillary SOL-INCA study (2016–2018).

Background

- The term "Alzheimer's disease" should be expressed in abbreviated form after the abbreviation.

Thank you for pointing this out. We have revised the manuscript to ensure that "Alzheimer's disease (AD)" is abbreviated upon first mention and that the abbreviation is used consistently thereafter.

Method

Data

- As mentioned in the Abstract section, it would be helpful for readers to clarify whether this study is cross-sectional or longitudinal.

Thank you for pointing this out. We have clarified in both the Abstract and Methods sections that this is a cross-sectional study based on data collected during Visit 2 of SOL-INCA (2016–2018).

- The authors did not mention whether individuals with any psychiatric or neurologic disorders that can affect the cognitive function were included.

Thank you for this comment. We have clarified in the Methods section that individuals were not excluded based on psychiatric or neurologic disorders. As this is a community-based study, the sample reflects a broad range of health conditions typical of the population. We now acknowledge this as a potential limitation in the Discussion, as such conditions may influence cognitive performance and biomarker levels.

- Clinical diagnosis of cognitive status (normal, MCI, or dementia) is also required ,if available.

Thank you for this suggestion. We have now included clinical cognitive status classifications (cognitively unimpaired vs. mild cognitive impairment) in the diagnostic criteria in the Methods section. While dementia diagnoses were not available for this visit, the classification of cognitive status was based on neuropsychological testing and established diagnostic protocols as previously published. We have also noted this in the Discussion and included stratified analyses by cognitive status in the Supplementary Materials

Blood collection and processing procedures

- "All coefficients of variability of the blinded duplicates were <11.3."  If "coefficients of variability" means "coefficients of variation," please correct the term with the appropriate unit for "11.3".

Thank you for pointing us to this issue. We have corrected the term from "coefficients of variability" to "coefficients of variation" and clarified the statement to read: "All coefficients of variation (CV) for blinded duplicate samples were <11.3%," which reflects the appropriate unit.

Covariates

- The term "high school" is expressed as "HS" in Table 1, but not in the manuscript.

Thank you for pointing this out. We have revised the manuscript to ensure consistent terminology by either spelling out "high school" or using the abbreviation "HS" consistently across the text and tables.

- The terms "CKD" and "BMI" are used in abbreviated form in the manuscript, but in full-term in the Table 2. Unifying the expression is recommended.

Thank you for the helpful suggestion. We have revised the manuscript and tables to ensure consistent use of abbreviations for "chronic kidney disease (CKD)" and "body mass index (BMI)" across the text and all tables. Full terms are now defined at first mention, and abbreviations are used consistently thereafter.

Analytic Approach

- "First, we characterized the SOL-INCA target population using the covariates included above (Table 1)." -> This sentence seems unnecessary in view of the redundant description at the beginning of the "Results" section.

Thank you for pointing this out. We agree that the sentence was redundant and have removed it to improve clarity and reduce repetition in the manuscript.

- "Extreme values (outliers at extreme values above and below the mean) of the plasma biomarkers were excluded..." -> Please clarify the criteria defining the extreme values.

Thank you for your comment. We have revised the manuscript to clarify that the primary analyses now use natural log-transformed plasma biomarker values to address skewness in their distributions, rather than excluding extreme values.

Results

- The title "Primary analysis" would be better replaced with a more appropriate title that describes the content of the analyses.

Thank you for the suggestion. We have revised the section title "Primary analysis" to a more descriptive title that reflects the content of the analyses. The new title is "Associations Between Plasma Biomarkers and Cognitive Performance."

- Table 2 is unnecessarily duplicated in page 21 and page 23.

Thank you for noting this formatting issue. We have removed the duplicate version of Table 2 to ensure the table appears only once in the manuscript.

- Page 9: The abbreviated term "CI" has been used as full term (confidence interval) in the Method section and last part of page 9.

Thank you for pointing this out. We have reviewed the manuscript and ensured consistent use of the abbreviation "CI" for "confidence interval" after its first definition. Instances where both forms were used inconsistently have been corrected.

Discussion

- Please discuss more thoroughly the issue about lower effect size of Abeta42/40. It could be result from the cross-sectional nature of this study, based on the amyloid cascade hypothesis, or the assay's issue (Simoa technique).

Thank you for this important suggestion. We have expanded the Discussion to more thoroughly address the lower estimates observed for the A β 42/40 ratio. Specifically, we note that the smaller regression estimates may reflect the cross-sectional nature of the study, as the amyloid cascade hypothesis suggests that amyloid pathology accumulates over time, often preceding detectable cognitive changes. Additionally, we acknowledge that measurement limitations related to the Simoa assay, such as sensitivity at lower concentrations and variability across plates, could have contributed to attenuated estimates. We now highlight these factors as important considerations when interpreting our findings.

- Individuals were from US metropolitan, which cannot represent the whole Hispanic/Latino group.

Thank you for this important point. We have added a statement to the Discussion noting that participants were recruited from four U.S. metropolitan areas, which may limit the generalizability of our findings.

Reviewer #2 (Remarks to the Author):

In this study, Márquez et al. investigate the associations between plasma biomarkers for AD and cognitive performance in a large population (N = 5730) of Hispanic/Latino individuals. They show that in this sub-population, plasma biomarkers of AD are associated with various aspects of cognitive performance, but that these were differentially affected when controlling for several demographic factors and comorbidities. The study is important as data regarding plasma biomarkers for AD in Hispanic populations is lacking, especially in such a large sample. Moreover, the manuscript is really well written. However, there are some statistical concerns and additional analyses to be addressed.

Major comments

Methods

1. Line 123: how were the z-scores derived for the global cognitive composite score? Were they based on cognitively unimpaired individuals? Please add a brief description.

Thank you for this important question. We have added a description to the Methods section clarifying that cognitive test scores were standardized into z-scores using the means and standard deviations of the SOL-INCA target population, not limited to cognitively unimpaired individuals. These standardized scores were then averaged to create a global cognitive composite score. We note this approach allows for population-based interpretation of cognitive performance relative to the study target population.

2. Line 124: “cognitive tests used at Visit 2”, does that mean that different tests were used for the visit 2 data? Were visit 1 and visit 2 z-scores used interchangeably? Please clarify in the manuscript to avoid confusion.

Thank you for the opportunity to clarify. We have revised the manuscript to state that all cognitive analyses in this study were based solely on cognitive test data collected at SOL-INCA. Although cognitive testing was also conducted at Visit 1, Visit 1 data were not used in the current cross-sectional analyses, and no z-scores were derived from Visit 1 performance. This clarification has been added to the Methods section to avoid confusion.

3. The statistics should be described more in depth. Several things are missing:

- a. Line 175 mentions that outliers were defined as extreme values, but how were these extreme values defined?
- b. Did you address the regression assumptions for the linear regression models? I.e., were biomarker levels log-transformed, was normality checked, etc.? Please add to the manuscript. Related to this: in Figure 1 it looks like plasma biomarker levels were also z-scored, but this is nowhere explained. Please explain how z-scoring was done for the biomarkers.

A. Thank you for pointing this out. We have since revised our analytic approach and now use natural log-transformed plasma biomarker concentrations in the primary analyses to account for skewness and reduce the influence of outliers. This approach is described in the Statistical Analysis section of the Methods. Sensitivity analyses excluding values above or below 3 SD from the mean were retained as supplementary analyses to assess robustness of findings.

B. Thank you for this important point. It is of importance to note that some of the regression assumptions do not directly apply when using complex survey designs (i.e., clustering, stratification) and that these models by design are robust to known violations of IID. All analyses use the survey functionalities of Stata. We have added language to the manuscript to explain this. In the revised manuscript, we apply log-transformation to the plasma biomarker values to reduce influence of the right-skewed distributions.

4. A lot of models are being executed here but nothing is mentioned about correcting for multiple comparisons. Hence, I would suggest correcting for multiple comparisons here per cognitive test per model (so within your Global Cognition models, correct for the 4 biomarker models you conduct within M1, and then for the 4 within M2, etc.), which will probably mean that some of the weaker p-values will become insignificant.

Thank you for raising this important point. We acknowledge the potential for type I error given the number of statistical tests conducted. However, we chose not to formally apply corrections for multiple comparisons, such as Bonferroni adjustment, because the biomarkers were selected *a priori* based on strong theoretical and empirical rationale. We have added a statement in the Discussion to acknowledge this limitation and encourage cautious interpretation of marginal p-values.

Results

1. Please add correlation plots between cognitive performance and raw plasma biomarker levels (e.g. to supplement, or as main figure 1).

Thank you for the suggestion. In response, we have added correlation plots between cognitive performance scores and log-transformed plasma biomarker levels to the Supplementary Materials. These plots help visualize the linear associations between the biomarkers and cognitive outcomes and complement the regression analyses presented in the main results.

2. Please add a M0: show the effects of just age, sex, education, and Hispanic/Latino background on cognition. Perhaps these already contribute quite a lot to differences in cognition?

Thank you for this suggestion. We added a (M0) that is unadjusted and a new M1 model that adjusts only for age, given its strong influence on both plasma biomarker levels and cognitive performance. Our goal was to isolate and highlight the effects of age in this population, particularly given the wide age range (50–86 years). We describe these models in the Methods section and include their results in the updated figure and Results text. Models adjusting for additional sociodemographic variables (e.g., sex, education, background, and field center) remain in M2 and beyond.

3. You could consider displaying the models in Figure 1 in a forest plot rather than as linear models and show the beta coefficient of the biomarker for each model. That would show nicely how the beta changes between models.

Thank you for this helpful suggestion. We agree that a forest plot offers a clear and concise way to visualize changes in beta coefficients across models. We have revised Figure 1 to display the regression coefficients for each biomarker across models using a

forest plot format. This updated visualization enhances interpretability by illustrating how the associations change with progressive adjustment for covariates.

4. Please report R² for the models in addition to std. B (95% CI). To see whether one model is significantly better than the other, I'd suggest comparing the R²'s between models with a bootstrap-hypothesis test for instance, or to compare models based on their AIC.

Thank you for this suggestion. While we agree that comparing R² values or AIC across models can offer insight into model fit, we believe that such model comparison analyses—particularly using bootstrap hypothesis testing (not directly applicable when using data from complex survey design with methods that adjust for clustering and stratification as well as unequal probability weighting of observations)—are beyond the scope of the current study. Our primary focus was on estimating associations between plasma biomarkers and cognitive performance across progressively adjusted models, rather than evaluating model selection or fit in detail. We have opted to present regression coefficients and confidence intervals to illustrate the pattern and strength of associations.

5. Please stratify analyses between CU and CI individuals, if you have cognitive status data available (unclear whether that is the case). This would be valuable information. Please also add cognitive status as a variable to Table 1, and add a supplementary table split between cognitive groups, if sample sizes allow it.

Thank you for this excellent suggestion. Cognitive status data were available for our sample, and we have now added a descriptives table split by this variable (cognitively unimpaired vs. mild cognitive impairment) in supplement. In addition, we conducted stratified analyses by cognitive status and present the results in a new table. These stratified results provide additional context regarding how associations between plasma biomarkers and cognitive performance may differ by clinical cognitive status. We also reference these findings in the Results and Discussion sections.

6. With your large sample size, it would be interesting to see what happens when adding all biomarkers to the same model with cognition as outcome and see if they all remain significantly associated. You should have sufficient power to try this out! Can be a supplementary figure/table.

Thank you for this insightful suggestion. We conducted an additional analysis in which all plasma biomarkers were entered simultaneously into the same model with cognitive performance as the outcome. The results are presented in a new supplementary table. This approach allowed us to assess the independent contribution of each biomarker while accounting for shared variance. We reference this supplementary analysis in the Results and Discussion sections.

Minor comments

1. Please mention the sample size in the abstract.

Thank you for the suggestion. We have revised the Abstract to include the sample size. We also provide a detailed chart of the sample inclusion/exclusion in supplement.

2. Table 1: I would suggest formatting the demographics for categorical variables as n (%), this is easier to interpret and highlights your large sample size (which is a strength!).

Thank you for this helpful suggestion. We have reformatted the demographic data for categorical variables in Table 1 to display results in accordance with standards of presentation for complex design data.

3. Line 61: these non-specific biomarkers are involved in both AD and other brain diseases, please correct.

Thank you for this observation. We have revised the wording in the manuscript to clarify that these are non-specific biomarkers involved in both AD and other brain diseases.

4. Line 76: please clarify in what way the APOE-e4 distribution varies among Hispanics/Latinos, and why that is relevant. Also in line 277: why is the effect of APOE-e4 especially relevant here?

Thank you for this comment. We have clarified in the manuscript that the distribution of the APOE-e4 allele varies among Hispanic/Latino subgroups, with a lower prevalence observed in individuals with higher Amerindian ancestry compared to those with European or African ancestry. This variation is relevant because it may influence Alzheimer's disease risk and modify the relationship between APOE-e4 status, plasma biomarkers, and cognitive outcomes in this population.

5. Please add a section to the discussion regarding the small beta's for the plasma biomarkers. The effects of the plasma biomarkers on cognition seem to be rather small.

Thank you for this thoughtful comment. We have added a section to the Discussion noting that the observed associations between plasma biomarkers and cognitive performance were statistically significant but small in magnitude. We now highlight that these modest effect sizes may reflect the early or preclinical nature of biomarker changes in cognitively unimpaired populations, the influence of unmeasured confounding factors, and the need for longitudinal follow-up to assess predictive value over time.

6. Please add to the limitations that your sample consists of self-identified Latino's which can introduce some bias

Thank you for this important point. We have added a statement to the Discussion acknowledging that our study includes self-identified Hispanic/Latino individuals, which may introduce some degree of bias due to heterogeneity of ancestry and culture within this population.

Reviewer #3 (Remarks to the Author):

Paper # COMMSMED-25-0342-T

Thank you for the opportunity to review this manuscript by Márquez et al., which describes the association between AD plasma biomarkers and cognitive performance in a large cohort of US Latinos. Blood biomarkers have emerged as accessible and highly promising tools for advancing the diagnostics of AD. However, more data on how these biomarkers may vary across diverse ancestries is critical in order to ensure broad and equitable translation to clinical practice.

The manuscript has some weakness in the presentation and discussion of the results. Specifically, I have the following concerns that need to be addressed.

Background.

Lines 65-66: the sentence is not supported by a pertinent citation. In addition, I suggest to briefly describe the literature data on the validity of blood biomarkers in clinical practice.

We thank the reviewer for this suggestion. We have revised the sentence to include appropriate citations and a brief summary of the growing body of evidence supporting the clinical validity of blood-based biomarkers for Alzheimer's disease. Specifically, we now reference key studies demonstrating the strong correlation of plasma biomarkers (e.g., pTau-181, NfL, A β 42/40) with established CSF and PET imaging biomarkers, their discriminative accuracy for AD versus other neurodegenerative conditions, and their potential utility in both research and clinical contexts.

Lines 79-81: I suggest to add a brief description of the literature data on blood biomarkers in Hispanic/Latino populations, clarifying the gap that the present study would fill in.

Thank you for this helpful suggestion. We have added a brief description in the Background section highlighting that existing studies on plasma AD biomarkers have largely been conducted in non-Hispanic White populations. Only a limited number of studies have examined these biomarkers in Hispanic/Latino populations, and most have focused on smaller samples. Additionally, they have been conducted in specific subgroups—such as Caribbean Hispanics (e.g., WHICAP), Cuban Americans (e.g., 1Florida ADRC), and Mexican Americans (e.g., HABLE and HABS-HD). These subgroups may vary in ancestry and culture. Our study addresses this gap by examining biomarker-cognition associations in a large, community-based cohort of diverse Hispanic/Latino adults.

Lines 85-87: The pre-specified hypothesis is in line with what know on blood biomarkers in non-Hispanic white people. Did the Authors have hypotheses reflecting the peculiarity of the population under study?

Thank you for this comment. Our pre-specified hypothesis was primarily informed by prior research on blood biomarkers in non-Hispanic White populations, where elevated pTau-181, NfL, and GFAP—and lower A β 42/40—are associated with poorer cognitive outcomes. However, we also hypothesized that these associations may differ in Hispanic/Latino populations due to differences in cardiometabolic burden, kidney dysfunction, and APOE- ϵ 4 allele frequency. We have now clarified this in the Introduction to reflect both the alignment with existing literature and our interest in investigating population-specific patterns in this underrepresented group.

Minor:

Line 48: 'deposition' should be more appropriate than 'formation'.

Line 56: 'Associated' should be more appropriate than 'linked'.

Line 69: I would suggest to use the term 'Relevant/important' instead of 'a few'.

Thank you for these helpful wording suggestions. We have revised the manuscript accordingly:

- In line 48, "formation" has been replaced with "deposition" to more accurately reflect the underlying biological process.
- In line 56, "linked" has been changed to "associated" for greater clarity.
- In line 69, "a few" has been replaced with "important" to better convey the significance of the studies referenced.

Methods.

The information about the proportion of MCI is of great relevance and should be reported. According to González et al. *Alzheimers Dement* 2019, the data is available. The regression models should be run separately in cognitively intact and MCI populations to overcome the limitation recognized in the Discussion.

Thank you for this important suggestion. We have now included the proportion of individuals with mild cognitive impairment (MCI), based on cognitive status classifications available in the SOL-INCA dataset as described in González et al., *Alzheimer's & Dementia*, 2019. In addition, we conducted stratified regression analyses separately for cognitively unimpaired and MCI groups to evaluate whether the associations between plasma biomarkers and cognitive performance differ by cognitive status. These results are presented in a new supplementary table and referenced in the Results and Discussion sections.

Pag. 5, lines 107-109: for clarity, I suggest to move the information on excluded data in the pertinent paragraphs, i.e., blood collection (n=151, samples not collected), covariates (n=496) and statistical analysis (outliers). Please, specify what is meant by 'missing exposure' in supplemental figure 1.

Thank you for this helpful suggestion. We have revised the Methods section to report exclusions in the relevant subsections for clarity:

- In the blood collection paragraph, we now state that 151 participants were excluded due to unavailable or unprocessed samples.
- In the covariates paragraph, we specify that 496 participants were excluded due to missing covariate data.
- In the statistical analysis section, we clarify the number of participants excluded due to censored values (falling outside of the range of detection). In the revised manuscript, we do not remove outliers in the primary analysis. We log-transform the biomarkers to reduce skew.

Additionally, we updated the caption of Supplementary Figure 1 to clarify that "missing exposure" refers to missing plasma biomarker data due to unavailable or unprocessed samples.

Pag. 6, line 123: The Authors refer to a description of the z-score computation, which is not presented.

Thank you for noting this oversight. We have revised the Methods section to include a description of the z-score computation. Specifically, raw cognitive test scores were standardized into z-scores using the weighted mean and standard deviation of the SOL-INCA study target populations. These z-scores were then averaged to generate a global cognitive composite score.

Lines 161-162: the value of triglycerides used to determine the presence of dyslipidemia is not specified.

Thank you for pointing this out. We have revised the Methods section to specify that dyslipidemia was defined as LDL-cholesterol ≥ 160 mg/dL, HDL-cholesterol < 40 mg/dL, or triglycerides ≥ 200 mg/dL.

Lines 170-171: the reference to the table 1 is not appropriate. Herein the Authors should specify the statistics used to describe the population.

Thank you for this suggestion. We have revised the text to remove the inappropriate reference to Table 1 and we report means and standard deviations for continuous variables with numbers and percentages for categorical variables.

Line 173: the term 'survey' to define a regression model seems unusual. Please specify.

Thank you for this observation. We have revised the manuscript to clarify that we used survey-weighted linear regression models, which account for the complex sampling design of the HCHS/SOL study. The term "survey-weighted" reflects the use of probability weights, strata, and clustering in the regression models to obtain estimates generalizable to the target population.

Line 175: the analysis of outliers should be described in more detail (more than $n=?$ standard deviations above or below the mean).

Thank you for this helpful comment. In the revised manuscript, we do not remove outliers in the primary analysis. We log-transform the biomarkers to reduce skew.

The statistical software used to make the analyses should be reported.

Thank you for this suggestion. We have updated the Methods section to indicate that all analyses were conducted using Stata 17, which supports survey-weighted regression models appropriate for complex sampling designs. In addition, some of the figures presented in the manuscript were generated using R version 4.4.2.

Minor:

Line 168 should be eliminated.

Thank you for your suggestion. We have removed line 168 from the manuscript as recommended.

Results.

The raw data of neuropsychological test scores and blood biomarker values should be reported as supplementary tables, for completeness of the data.

Thank you for this helpful suggestion. We have added the raw distributions of neuropsychological test scores and plasma biomarker values (means and standard deviations) in Supplementary Table 1 to enhance transparency and completeness.

In addition, it would be very interesting to show the mean levels of each of the biomarkers compared to non-Hispanic white individuals; indeed, the influence of ancestries on AD blood biomarkers is not fully understood (see doi: 10.1212/WNL.0000000000207675).

Thank you for this suggestion. We agree that comparing biomarker levels between Hispanic/Latino and non-Hispanic White individuals would provide important context, especially given emerging literature on ancestry-related differences in AD biomarkers. However, non-Hispanic White comparator data were not available within the SOL-INCA or HCHS/SOL studies. While we reference prior studies that have reported such differences, incorporating a direct comparison is beyond the scope of this analysis. We have added a note in the Discussion acknowledging this as an important direction for future research.

The table 2 should report number(%). Age should be reported at the top of the table together with other sociodemographic features. In order to lighten the table, mean BMI and non-APOE4 allele frequencies can be moved in the text.

Thank you for the helpful suggestions. We have revised Table 1 to report categorical variables as number and percentage (n [%]) and moved age to the top of the table alongside other sociodemographic variables. Given the relevance of BMI and APOE-ε4 status as covariates in our adjusted models, we have retained these variables in Table 1 to provide important context for the analytic sample.

Lines 192-216: the paragraph is very difficult to read. The Authors should describe the results for each biomarker in separate paragraphs. In addition, the information in the text should not duplicate that reported in the table 2. In the text, the Authors should highlight the most significant results (e.g., the associations that remain significant in the full adjusted model).

Thank you for this helpful feedback. We have rewritten the Results section to present findings for each biomarker and focused the text on highlighting the most significant associations that remained in the fully adjusted models. Redundant reporting of values already presented in Table 2 has been minimized to improve readability and clarity.

The figure 2 is hard to read: the models from M2 to M5 are overlapped.

Thank you for your feedback. To improve readability and address overlapping estimates, we have revised Figure 1 to display the results as a forest plot. This format more clearly illustrates the regression coefficients and confidence intervals across models (M0–M5)

for each biomarker, making it easier to compare the magnitude and direction of associations.

Discussion.

Lines 228-230: the term 'attenuated' seems incorrect: the associations are not significant in adjusted models. The Authors should summarize the results in a more coherent and concise manner, e.g., first reporting significant associations with global cognition, and then detailing which cognitive domain the association is driven by, for each blood biomarker.

We thank the reviewer for this helpful suggestion. We have revised the relevant section of the Results to improve clarity and logical flow. We removed the term "attenuated" in cases where associations were not statistically significant after adjustment and now summarize results more concisely, beginning with associations with global cognition followed by domain-specific findings for each biomarker.

Lines 249-254: these considerations have been already reported in the Introduction. They can be omitted.

Thank you for your comment. We have revised the manuscript to remove the redundant considerations in Lines 249–254, which were already presented in the Introduction.

Line 267: the term 'cognitive outcome' is appropriate for clinical trials, not population-based studies.

Thank you for this suggestion. We have revised the wording to replace "cognitive outcome" with "cognitive performance," which is more appropriate for the context of our population-based study.

Lines 269-270: Has a measure of cognitive reserve been collected? What 'environmental influences' refer to? Please clarify.

Thank you for this observation. In our discussion, the mention of "environmental influences" was intended to refer broadly to factors such as educational attainment, socioeconomic status, health care access, and sociocultural experiences that may contribute to variability in cognitive performance. However, we have removed the term "environmental influences" from the manuscript to improve clarity and maintain focus. This change ensures that the discussion remains grounded in the variables directly assessed in the study.

Lines 270-275: the hypothesis that the association between lower plasma Aβ42/40 and reduced verbal fluency can be underlined by CCA is interesting but quite speculative. This should be made clearer. To my best knowledge, verbal fluency do not rely on processing speed. A reference for this statement is not provided.

Thank you for this insightful comment. We agree that the proposed link between lower plasma Aβ42/40 and reduced verbal fluency via underlying cerebral amyloid angiopathy (CAA) is speculative and have revised the text to reflect this more clearly. We now present this as a potential hypothesis for future research rather than a definitive interpretation. Additionally, we have removed the reference to processing speed in relation to verbal fluency, as this connection was not directly supported by existing literature.

Lines 287-288: this sentence seems to contradict the following one: neuroinflammation can disrupt circuits essential for executive functioning, but GFAP, a marker of inflammation, is associated with memory, not executive functions.

Thank you for this thoughtful observation. We agree that the original phrasing created a contradiction. We have revised the sentence. In our study, GFAP was more consistently associated with learning and memory-related outcomes in older adults and in those with MCI. This suggests that the effects of neuroinflammation may be domain-specific or context-dependent and warrants further investigation.

Lines 290-291: this sentence is quite general and it is not supported by a citation. The implication of the results for clinical practice should be addressed.

Thank you for this helpful comment. We have revised the sentence to make it more specific and added appropriate citations to support the statement.

Reviewer #4 (Remarks to the Author):

The manuscript titled “Plasma biomarkers of Alzheimer’s disease and related dementias and cognitive performance among Hispanics/Latinos: Findings from the HCHS/SOL and SOL-INCA” explores the relation between neurodegeneration and Alzheimer’s disease plasma biomarkers with cognition scores in a cohort of Hispanic and Latino people living in multiple US cities. The measurements are taken on the ultrasensitive SIMOA platform and on an impressive amount of patients, >5700, and they are also well characterized with extensive information on each patient of both a socioeconomic and health nature. The aim of the study is clearly stated in the abstract and introduction and the authors do a good job in not digressing from the aim and fulfilling it, as they show some associations between the plasma biomarker levels and the cognitive scores. The manuscript is well written and the introduction and methods are clear to follow. Overall, the article could contribute to the field by reporting measurements on a population that has been previously under-represented in these biomarker studies.

Major comments:

1. The manuscript’s strength of a very high number of participants, comes with statistical challenges that were not addressed. All 5700+ patients were treated as one group, this is somewhat unusual, since often large cohorts are segmented in subgroups, either diagnostic or demographic ex. age brackets. This approach would also reduce the chances of having significant P-values that are not clinically meaningful. The reported Beta coefficients and confidence intervals in Table2 are statistically significant but very close to 0. Have the authors considered dividing the population in groups or to validate the associations in a subset of patients?

Thank you for this thoughtful comment. We agree that stratifying the analysis can provide more nuanced insight, particularly given the large sample size and potential for statistically significant but clinically small associations. We have added stratified analyses by age group (<60, 60–69, and ≥70 years) and by cognitive status (cognitively unimpaired vs. mild cognitive impairment). These subgroup analyses are now included, and relevant findings are referenced in the Results and Discussion sections. This additional stratification allows us to explore whether biomarker associations with cognition differ meaningfully across age and cognitive status.

2. No raw data is shown in the manuscript. The only figure shows estimates from the model. Some correlations between marker and score or violin plots comparing some groups (ex. High-scoring Vs. low-scoring) would be beneficial to further show if there is a relation between the cognitive scores and plasma biomarker levels and their distribution.

Thank you for this suggestion. To improve transparency and data visualization, we have added supplementary figures showing the distributions of plasma biomarker levels and cognitive scores. In particular, we included scatterplots with trend lines to illustrate the bivariate correlations between each biomarker and cognitive performance. These visualizations complement the regression results and provide a clearer view of the underlying data patterns.

3. The rationale for the choices of the covariates for the model is not argued clearly. In Figure 1, it is possible to see how models 2-5 are basically identical in performance. I would suspect that this is driven by adjusting for age, which drives most of the effect, especially considering the

wide-spread age span (50-86) present in the samples. It is known how biomarkers, for example NfL (PMID: 35865350) and other biomarkers used in the study (PMID: 37237388), are positively correlated with age. Can the authors clarify if they have used other combinations of models and if a model just adjusted by age would perform the same as M2-M5?

We thank the reviewer for this insightful comment. We agree that age is a major driver of both biomarker levels and cognitive performance, and its inclusion markedly attenuates the observed associations. This is evident in our modeling progression, where the largest change in effect sizes occurred between the unadjusted (M0) and age-adjusted (M1) models, with subsequent models (M2–M5) showing relatively minor incremental changes.

To clarify, the covariates included in M2–M5 were selected a priori based on existing literature and their known or hypothesized relevance in both Alzheimer’s disease biomarker expression and cognitive outcomes, particularly in Hispanic/Latino populations (e.g., cardiometabolic risk factors, kidney function, and APOE ϵ 4 genotype). While the additional adjustments did not substantially alter effect estimates beyond age, we retained these covariates to account for potential confounding and to enhance comparability with prior studies using HCHS/SOL and SOL-INCA data.

As requested, we further examined models including only age adjustment (M1) and compared them directly with M2–M5. The results confirm that age is the primary driver of the attenuation, and the additional covariates do not significantly change model estimates. We have clarified this point in the Discussion section and included references (PMID: 35865350; 37237388) supporting the strong correlation between age and biomarkers such as NfL.

5. For such a large study run on the SIMOA platform (an estimation of >70 plates per assay), one can expect a technical variation of at least around 10%. The manuscript should include whether any control samples were used in the plates to keep track of such variation. If they have been used the values for the repeatability and precision should be included in the manuscript and whether the data has been normalized based on the controls to reduce this variability. Or have any other statistical strategies been used to address this? If they have not been included, this should be acknowledged as a limitation of the study.

Thank you for this important and detailed comment regarding assay precision and quality control procedures.

In response, we have revised the Methods section to address plate-to-plate variability and clarify the quality control measures implemented for the plasma biomarker assays run on the Simoa platform. Specifically:

- We included pooled control samples in every assay plate for GFAP, NfL, and pTau-181. These controls, derived from a pool of 30 anonymized donors, demonstrated coefficients of variation (CVs) below 12% across plates, indicating good repeatability and precision for these analytes. Importantly, we did not observe any significant systematic laboratory drift or shift in the assay performance across the plates and the variation was consistent

with random variation across plates. The lack of systematic bias in assay performance across plates suggests that statistical normalization across plates is not required in this study.

- For A β 40 and A β 42, the pooled control yielded very low concentrations, resulting in high variability (CV ~50%). Because of this limitation, we relied on the manufacturer-provided synthetic controls and blinded duplicate samples to monitor precision for these two analytes.
- We did not apply statistical normalization across plates using these controls; however, we monitored assay performance through CVs and blinded duplicates. We now acknowledge this in the manuscript and have added a limitation noting that plate-to-plate variation was not adjusted for statistically, which may contribute to measurement error, particularly for A β 42/40.

We appreciate the reviewer's thoughtful suggestion, which helped improve transparency around assay reliability and study limitations.

Discussion

4. A discussion between clinical and statistical significance is lacking in the limitations section of the manuscript. With such small Beta coefficients and such a large sample size it is likely the clinical significance is limited. Unless, the results can be replicated in a smaller subset of samples.

We appreciate the reviewer's observation regarding the distinction between statistical and clinical significance. We have now added language to the Limitations section to explicitly address this issue. While the large sample size provided sufficient power to detect statistically significant associations, we agree that the observed beta coefficients were modest and likely reflect small effect sizes with limited clinical relevance at the individual level. These findings should therefore be interpreted as population-level associations that may reflect early or subclinical processes rather than diagnostic thresholds.

We also emphasize that our results support the potential of plasma biomarkers to detect subtle cognitive differences in large, heterogeneous community-based populations, but further research—including replication in smaller or clinically enriched samples—is needed to determine their prognostic and clinical utility. The manuscript has been revised accordingly. Thank you for highlighting this important distinction.

Minor comments:

1. Line 78-79: The summarization of REF.12 is not entirely faithful, as the manuscript does not argue that comparisons are not applicable between different socioeconomic backgrounds, rather that it might affect the risk-factors to develop diseases, such as dementia.

Thank you for this comment. To ensure clarity and focus, we have removed the sentence: *"Moreover, there is no consensus on whether biomarker level comparisons are applicable across individuals from different socioeconomic backgrounds."* This

revision avoids potential misinterpretation and keeps the discussion aligned with the intent of the cited literature.

2. Line 79-80: But has any similar study been carried out in cohorts representing other ethnicities, what were the findings? Will they differ between this Hispanic and Latino cohort?

We thank the reviewer for this comment. Emerging research suggests that biomarker levels and their associations with cognitive outcomes may vary by ancestry due to differences in vascular risk profiles, kidney dysfunction, APOE genotype frequencies, social determinants of health, and access to care. Our findings provide novel insights by characterizing these associations in a large, population-based cohort of diverse Hispanic/Latino adults. While general patterns of association appear consistent—such as elevated NfL and pTau-181 being linked to worse cognitive outcomes—our results underscore the importance of studying these biomarkers in diverse populations to identify potential differences in biomarker distribution, thresholds of clinical relevance, and modifying factors. We have added a brief discussion of these points to the Discussion section to highlight both the consistency with and need for broader population-based evidence.

3. Line 92: The general description of the cohort says it is composed of individuals ages 18-74, in the abstract however it says the study included participants aged between 50-86. Were there any participants older than 74 then?

We appreciate the reviewer's attention to this detail. The parent HCHS/SOL cohort enrolled participants aged 18–74 years at baseline (Visit 1; 2008–2011). The ancillary SOL-INCA study was conducted several years later (2016–2018) and included participants aged 50 years and older at Visit 2. As a result, some participants who were in their early 70s at baseline had aged into their 80s by the time of SOL-INCA participation. Therefore, yes, a subset of participants were older than 74 at the time of the ancillary study, which is reflected in the age range reported in the abstract (50–86 years). To avoid confusion, we have clarified this point in both the Methods section and the abstract.

4. Line 108- 109: The sentence could benefit from a clarification that also the number of samples measured for each assay is displayed in the Supp. Fig 1., as this information is lacking in the text.

Thank you for this helpful suggestion. We have revised the text to clarify that the number of samples measured for each assay is provided in Supplementary Figure 1. This clarification has been added to the Methods section to enhance transparency regarding sample availability.

5. Line 111: In the “Neuropsychological assessment” paragraph, it could be helpful to include if there is a range between which patients are classified as having cognitive deficiencies, this would give more context later on to the estimates from the model, if for example a change of 1 in the score is meaningful or one of 10 is.

We thank the reviewer for this suggestion. In our analysis, we standardized each of the neuropsychological test scores using z-score transformation (mean = 0, standard deviation = 1) based on the analytic sample. As a result, all effect estimates can be interpreted in units of standard deviations of the cognitive outcomes. While a clinical classification threshold was not applied to define cognitive impairment on each test, the use of z-scores allows for comparability across domains and reflects relative cognitive performance within the study sample. We have added language in the Methods section (“Neuropsychological assessment”) to clarify the use of z-scores and the interpretation of the beta coefficients.

6. Line 174-177: The specific rationale of the definition of outlier is not clearly identified, other than a vague statement of “(outliers at extreme values above and below the mean)”. The statistical rationale should be stated ex. $\pm 3SD$ from the mean. Have the authors considered whether the upper outliers could be individuals with Alzheimer’s disease? Moreover, based on the numbers reported of outliers in Supp. Fig 1, their effect against 5700+ “non-outliers” is potentially negligible, has the analysis been conducted with and without outliers?

Thank the reviewer for this detailed and constructive comment. We revised our analytic approach to use natural log-transformed plasma biomarker concentrations in the primary analyses, rather than excluding extreme values. This transformation addresses skewness in the biomarker distributions while retaining all eligible participants. We now clarify in the Methods that values ± 3 standard deviations from the mean is used in a sensitivity analysis (not the primary analysis), and we have updated the relevant text and Supplementary Table accordingly. Results from models excluding these extreme values were consistent with the log-transformed models, supporting the robustness of our findings. These revisions have been made in the Statistical Analysis and Sensitivity Analysis sections.

We agree that elevated biomarker levels among upper outliers could reflect underlying neurodegenerative disease, including undiagnosed Alzheimer’s disease. However, given the population-based nature of the cohort and lack of clinical diagnoses for all participants, we were cautious in interpreting outliers as AD-specific without supporting clinical data.

7. Line 177: The authors should state in the text what the “outcome” is and also clearly state what the predictor variables are, this would also improve the clarity of Figure 1.

Thank you for this helpful suggestion. We have revised the manuscript to clearly define cognitive performance as the primary outcome variable and plasma biomarker concentrations ($A\beta_{42/40}$, pTau-181, NfL, GFAP) as the main predictors. We also updated the caption and labeling of Figure 1 to improve clarity regarding the directionality of the associations being modeled.

8. Line 181- 183: In Figure 1 it is not clear where the implementation of the post-hoc ANOVA is, moreover was the post-hoc ANOVA done to compare the models? Was there any significant change between them? Where is this ANOVA outcome reported? Even if the result is not significant it should be stated in the results.

Thank you for noting this oversight – we had estimated and plotted average marginal means of the cognitive scores across the continuum of the z-scored plasma biomarkers. We now present the forest plots to visualize beta coefficients across in place for the marginal means plots, and the text in the manuscript has been updated.

9. Line 191: The section titled “Primary analysis” would benefit from referring to the models by their name ex. M1, M2 etc. Otherwise, for example in line 196, it is not immediately clear what the “sociodemographic covariates” are.

Thank you for this helpful suggestion. We have revised the “Primary analysis” section to refer explicitly to each model by name (e.g., M1, M2) throughout the text.

10. Line 199: In table 2 the beta coefficients are reported as being standardized, it would help the clarity to state it also in the main text and label the β accordingly.

Thank you for this helpful suggestion. We have revised the manuscript to improve clarity regarding the nature of the beta coefficients in Table 2. Specifically, we now clearly state in the Results section that the beta coefficients are unstandardized, consistent with Table 2. We have also ensured that the table caption and figure titles are labeled accordingly to reflect that the estimates are based on log-transformed but unstandardized plasma biomarker values. This clarification has been made to avoid any ambiguity and to ensure alignment between the text and the tables/figures.

11. Line 251-254: The makeup of the Hispanic/Latino group is described as being very heterogenous, could the authors elaborate on why they did not check if there are differences between the various Hispanic/Latino ancestries present in the cohort? The information is reported as being available in Table 1.

Thank you for this comment. We agree that examining differences in associations across Hispanic/Latino ancestry groups is an important and valuable direction for future research. While the cohort includes participants from diverse backgrounds, our primary aim in this manuscript was to evaluate overall associations between plasma biomarkers and cognition in a population-based Hispanic/Latino sample. Stratified analyses by Hispanic/Latino background would substantially increase model complexity and reduce power within subgroup strata, particularly for biomarkers with more limited availability. However, we now acknowledge this as a limitation and have added a note in the Discussion highlighting the need for ancestry-specific analyses in future work.

12. Line 264-265: “it is unclear if amyloid-lowering drugs would benefit all populations at risk.” Please provide a reference for this statement or rephrase.

Thank you for the comment. In response, we have revised the manuscript to omit the sentence, “*it is unclear if amyloid-lowering drugs would benefit all populations at risk,*” to improve focus and avoid introducing unsupported speculation. We agree that without direct supporting evidence, this statement may be too broad for inclusion.

13. Line 266 & 283: the description of the markers as “non-specific” should be rephrased as “markers of neurodegeneration”.

Thank you for this important observation. While we initially described NfL and GFAP as “non-specific” biomarkers, we recognize the need for greater precision. According to the NIA-AA Research Framework, these markers reflect neurodegeneration and glial activation, which are involved in—but not specific to—AD pathophysiology. So, describing them as “non-specific biomarkers involved in AD pathophysiology” or markers of “neurodegeneration” and “neuroinflammation”, respectively, is more accurate and aligned with the current framework.

14. Line 277-278: Here APOE genotype is described as attenuating only slightly the relationship between GFAP and cognitive specific performances. Is this a reference to results from Model 5? In this case APOE is added after having adjusted for many covariates, have the authors checked if just adjusting for APOE status (perhaps along with age and sex) also attenuates the association only mildly? So, to increase clarity, the authors should state if it is M4 Vs. M5 or some other comparison.

We appreciate the reviewer’s comment. We have clarified in the revised manuscript that the reported findings are based on the fully adjusted models (Model 5), and that adjustments for sociodemographic factors, health characteristics, and APOE ϵ 4 genotype did not substantially attenuate the associations beyond the effect of age. Specifically, we now state that the largest attenuation occurred with age adjustment (Model 1), while subsequent adjustments for sociodemographic variables, health-related factors, and APOE ϵ 4 genotype (Models 2–5) led to relatively modest changes in effect estimates. This pattern underscores the dominant role of age in the observed associations and the robustness of our findings to additional covariate adjustment. We have revised the Results and Discussion sections to better reflect this modeling strategy and to improve clarity.

15. Line 288-290: It is not clear if the association between learning and GFAP is supported by evidence from this study or from data present in REF. 31. Please clarify this sentence.

We thank the reviewer for this comment. To improve clarity and maintain focus on the findings from our own study, we have removed the sentence in question and the corresponding reference (Ref. 31).

16. Line 295-296: Could the authors elaborate on why they speculate that “...it is possible that other non-AD prevention and treatment strategies should be sought for Hispanics/Latinos”. Do they intend “other” as in different from other ethnicities or as in lifestyle-changes that are non-specific to AD, ex. Weight loss. And are these specific to just Hispanics/Latinos or could benefit anyone regardless of ethnicity?

We appreciate the reviewer’s comment and agree that the sentence in question was speculative and potentially unclear. To improve focus and clarity, we have removed this sentence from the revised manuscript. We now concentrate our discussion on evidence-based findings from our analyses and avoid speculative interpretations regarding prevention strategies.

17. Line 304-305: If the information regarding patients being MCI or CU is available for participants it should be included in the demographics table. Moreover, how come this important metric was not used in any analysis? Was the MCI status determined with the cognitive tests

presented in this manuscript? If so, it would be beneficial to include the cut-offs in the methods section.

Thank you for this important comment. Cognitive status information (cognitively unimpaired [CU] vs. mild cognitive impairment [MCI]) was available for participants and has now been added to the demographics table (Supplementary Table 1). MCI classification was determined using the neuropsychological tests described in this manuscript and based on previously published diagnostic criteria (González et al., Alzheimer's Dement 2019). We have incorporated stratified analyses by cognitive status (CU vs. MCI) in the Supplementary Materials and referenced these results in the Results and Discussion sections.

18. Line 306: "...the influence of blood-based biomarkers on cognitive performance..." The sentence should be improved in the wording as the blood-based biomarkers do not influence cognitive performance.

Thank you for pointing this out. We agree that the original phrasing may have implied a causal relationship. We have revised the sentence to reflect the observational nature of the study more accurately.

19. Line 308-311: Whilst a large spread of ages could be a very interesting proposal and strength of this cohort, in this manuscript any differences regarding age are not presented. Have the population levels of the biomarkers been investigated at different age brackets? Currently, age seems to be included only as a variable in the models, which is therefore used to cancel the effect of age on the results.

Thank you for this thoughtful comment. In response, we conducted additional analyses stratifying participants into age brackets (<60, 60–69, and ≥70 years) to examine how plasma biomarker levels and their associations with cognitive performance may differ across the age spectrum. These stratified results are presented in a new supplementary table and referenced in the Results and Discussion sections. While age was included as a covariate in the main regression models, these additional analyses allowed us to explore age-related patterns without adjusting away their effects.

20. Figure 1: In the figure legend, it should be made clear which is the outcome variable and which is the predictor variable. Are we seeing the predicted global cognition scores based on biomarker levels or the other way around?

Thank you for this suggestion. In Figure 1, we now present forest plots visualizing coefficients, and we have included labels to clearly state that plasma biomarkers are the predictor variables and global cognitive performance is the outcome. This clarifies the direction of the associations modeled in the figure.

21. Figure 1: In the figure panels, the lines include some dots, what do the dots on the lines represent? In the analysis has the global cognition score been used as a numerical variable or ordinal categorical? Could the authors please clarify.

Thank you for your comment. We have updated Figure 1 to display the results as a forest plot for improved clarity. In the revised figure, each dot represents the beta coefficient from a regression model, and the accompanying horizontal line indicates the 95% confidence interval. Global cognitive performance was modeled as a continuous

outcome variable (composite z-score), not as an ordinal or categorical variable. This clarification has been added to both the figure legend and the Methods section.

22. Table 1: In the demographics table, age is only presented as the mean with SD. It would be beneficial to the reader to have it presented also in age brackets (ex. 50-60, 60-70 etc.) so have an idea of the distribution. Similarly, the amount of patients classified as with normal cognition or mild cognitive impairment should be included. As these two metrics are correlated with the levels of the biomarkers measured.

Thank you for this suggestion. We have revised the demographics table (Table 1) to include age categorized into brackets (50–59, 60–69, and ≥ 70 years) to better reflect the age distribution of the sample. Additionally, we have added a demographics table (Supplementary Table 1) by the cognitive status classification (cognitively unimpaired vs. mild cognitive impairment), as this variable is relevant to both cognitive outcomes and biomarker levels. These changes provide clearer context for interpreting the results.

23. Table 2: In the table please adjust the rounding in the cases where -0.00 is used it should read -0.01. This was also present in the manuscript.

Thank you for noting this. Since the value was closer to -0.001 than -0.01 for those cases, we have added a note to indicate that “Values near zero may appear as ± 0.00 ”.

Reviewer #5 (Remarks to the Author):

Dear dr. Marquez,

First, I would like to commend you for conducting research on minority populations, who represent a large and frequently understudied group affected by dementia. Your use of a substantial sample size and high-quality Simoa-based measurements underscores the methodological rigor of your work.

I understand your work to focus on:

Investigating associations between AD-specific (A β 42/40, pTau181) and non-AD specific (NfL, GFAP) plasma biomarkers and cognitive performance in a diverse cohort of community-dwelling Hispanic/Latino adults in the USA.

Although the manuscript's title initially piqued my interest, I must note that the content did not fully align with my expectations. Below are my major and minor concerns:

MAJOR CONCERNS

BACKGROUND

- Additional references would strengthen certain claims. For instance, the sentence: "More recently, the ATN framework has been implemented using more accessible and less expensive blood-based biomarkers ..." cites only one source.

Thank you for your suggestion. We agree that additional references would enhance the strength and credibility of this statement. In the revised manuscript, we have added supporting citations to reflect the growing body of literature on the implementation of the ATN framework using blood-based biomarkers. These include studies that validate the use of plasma biomarkers (e.g., A β 42/40, pTau-181, NfL, GFAP) within the ATN framework and their application in both research and clinical settings. The sentence has been updated accordingly to reflect this broader evidence base.

- You make mention of criteria for diagnosing AD, but leave out in the 'while alive' and in A-T-N framework (as diagnosing AD postmortem is still the golden standard, but clearly does not need to make use of blood-based biomarkers).

Thank you for this important point. We agree that clarifying the distinction between postmortem and *in vivo* diagnostic criteria strengthens the accuracy of our framing. We have revised the relevant section of the manuscript to specify that the ATN framework was developed to facilitate *in vivo* classification of Alzheimer's disease (AD) pathology while individuals are still alive. This revision provides more accurate context for the clinical and research utility of plasma biomarkers.

- In the statement: "3) biomarkers of common non-AD co-pathologies," it would be helpful to specify which co-pathologies you are referring to.

Thank you for your comment. In response, we have removed the phrase "biomarkers of common non-AD co-pathologies" to improve clarity and maintain focus.

- You note that “(3) the APOE-ε4 allele distribution varies among Hispanics/Latinos,” but do not address this stratification in the discussion.

Thank you for your comment. We agree that the distribution of the APOE-ε4 allele across Hispanic/Latino background groups may have implications for Alzheimer’s disease risk and biomarker interpretation. While we noted this variability in the Introduction, we did not conduct stratified analyses by background group in the current study. We now clarify this limitation in the Discussion and emphasize that future studies are needed to investigate potential ancestry-related differences in plasma biomarker associations with cognitive outcomes.

- You state there is no consensus on whether biomarker comparisons apply across socioeconomic strata. This point distracts slightly from your main findings.

Thank you for this observation. We agree that the statement about socioeconomic strata, while important, may divert focus from the primary objectives of our study. In response, we have removed this point from the manuscript to maintain clarity and emphasize the main findings related to plasma biomarkers and cognitive performance in a diverse Hispanic/Latino population.

DATA

- The participants section is confusing. It is unclear whether this study includes only the SOL-INCA samples or HCHS/SOL. Consider omitting excessive detail about HCHS/SOL to avoid confusion.

Thank you for pointing this out. We have revised the Participants section to clarify that our analytic sample is derived from SOL-INCA, an ancillary study of HCHS/SOL. To improve clarity and reduce confusion, we have streamlined references to HCHS/SOL and focused the description on SOL-INCA, which provided the data used in the current analysis.

- “A total of n=496 participants were excluded due to missing covariates” should specify which covariates. Also clarify the timing and criteria for “exclusion of extreme values.”

Thank you for your comment. We have revised the manuscript to clarify the exclusion criteria. Specifically, we now specify that the 496 participants were excluded due to missing data on key covariates used in the adjusted models, including age, sex, education, Hispanic/Latino background, field center, BMI, diabetes status, hypertension status, dyslipidemia, chronic kidney disease (CKD) status, and APOE ε4 genotype.

We also updated Supplementary Figure 1 to display the number of participants missing for each covariate. In addition, we note that in the primary analyses, biomarker values were natural log-transformed to reduce skew and account for outliers, rather than being excluded. In sensitivity analyses (now supplemental), we reran our models excluding individuals above or below 3 standard deviations from the mean for each biomarker.

Your four neuropsychological tests might be insufficient to represent global cognitive function.

Thank you for your comment. We acknowledge that our cognitive battery, which includes verbal learning, verbal memory, verbal fluency, and processing speed/executive function, does not fully represent all domains of global cognitive function (e.g., attention, visuospatial abilities). However, the composite score derived from these validated measures is consistent with prior HCHS/SOL and SOL-INCA studies and conceptually similar to composite scores like the Preclinical Alzheimer's Cognitive Composite (PACC), which are widely used in AD research. We now clarify this in the manuscript and acknowledge the limitation in domain coverage in the Discussion.

- In a multicenter study, explain how plasma samples were assayed. Was everything run simultaneously on a single Simoa platform? You mention NfL measurements using two different kits (N4PE vs. NF-light advantage). Were bridging samples included, and how was agreement between assays established? Was plasma pTau181 measured in singlicate or duplicate?

Thank you for your comment. We have added a sensitivity analysis that focuses specifically on the subsample of participants whose NfL levels were measured using the multiplex (Neurology 4-Plex E Advantage) platform (Supplementary Table 5). We repeated the regression modeling for these participants and found that the associations between NfL and cognitive outcomes were consistent with those observed in the full sample, which included both multiplex and simplex assays.

These findings support the robustness of our primary results and suggest that assay type did not materially affect the observed associations. The results of this sensitivity analysis are now presented in the Supplementary Table 5, and a corresponding note has been added to the Results and Methods sections.

- Judging from Figure 1, only the unadjusted model (m1) differs meaningfully from all the adjusted ones. Adding three further models does not seem to contribute much if incremental differences are minimal and under-discussed.

Thank you for this observation. We agree that the largest change in estimates occurs between the unadjusted model (M0) and the age-adjusted model (M1), which highlights the strong influence of age on both plasma biomarkers and cognitive performance. While models M2–M5 result in smaller incremental changes, we retained them to reflect our prespecified analytic plan and to demonstrate the robustness of associations after adjusting for a comprehensive set of demographic and clinical factors. We have revised the Discussion to note the relative stability of effect estimates across models and to interpret this pattern more clearly.

- You mention these covariates were included based on the literature. Did you confirm their relevance in your own dataset?

Thank you for the suggestion. The covariates included in our models were selected based on their established relevance in the literature and prior use in HCHS/SOL and SOL-INCA analyses. While we did not conduct formal variable selection procedures in the current study, descriptive analyses and prior work within this cohort support their relevance to both biomarker levels and cognitive performance. We have clarified this point in the Methods section.

DISCUSSION

- Please clarify the statement: “In clinic-based studies, there is evidence that supports the risk-stratification capabilities and utility of blood-based biomarkers that detect abnormal protein accumulation in the brain” How does risk-stratification specifically relate to your findings?

We appreciate the reviewer’s comment regarding the clarity and relevance of the statement about risk stratification. To improve focus and maintain alignment with the objectives of our study, we have removed this sentence from the manuscript. This change helps make the discussion clearer and ensure that all content is directly connected to our findings.

- Overall, the discussion might be more focused if it aligned directly with the research question. Consider minimizing tangential points, such as the reference to cerebral amyloid angiopathy (CAA).

Thank you for this helpful suggestion. To improve focus and ensure the discussion aligns more directly with our research question, we have removed the reference to cerebral amyloid angiopathy (CAA). This revision allows us to concentrate more clearly on the primary findings and implications of the study.

MINOR CONCERNS

ABSTRACT

- Mention explicitly that your participants are community-dwelling Hispanic/Latino adults.

Thank you for the suggestion. We have revised the manuscript to explicitly state that participants were community-dwelling Hispanic/Latino adults.

- Indicate which cognitive tests were used.

Thank you for your suggestion. We have revised the abstract to explicitly indicate which cognitive tests were used to assess performance: the B-SEVLT for verbal learning, Word Fluency for verbal fluency, and Digit Symbol Substitution for processing speed/executive function, and a global cognitive composite.

- Avoid phrases like “worse global cognition” for cross-sectional data; “poorer global cognitive performance” is more precise.

Thank you for the suggestion. We have revised the manuscript to replace phrases such as “worse global cognition” with “poorer global cognitive performance” to reflect the cross-sectional nature of the data more accurately.

- The text says “Elevated plasma biomarker levels of A β and tau ... are associated with lower cognitive performance,” but it is actually lower A β 42/40 that associates with poorer performance.

Thank you for pointing this out. The text has been revised to clarify that *lower* A β 42/40, reflecting greater amyloid burden, is associated with poorer cognitive performance, consistent with established interpretations of this biomarker.

BACKGROUND

- Use abbreviations consistently (i.e., “Alzheimer’s disease” versus “AD,” and “blood-based biomarkers”).

Thank you for pointing this out. We have reviewed the manuscript and ensured consistent use of abbreviations throughout—for example, using “Alzheimer’s disease (AD)” after the first mention and consistently referring to “blood-based biomarkers” thereafter as appropriate.

DATA

- Include protocol numbers for the Institutional Review Boards.

The HCHS/SOL and the SOL-INCA studies were reviewed and approved by the Institutional Review Boards of the University of California San Diego (Study #803924) and all participating sites (University of North Carolina-Chapel Hill Single IRB: #20-1900).

RESULTS

- The interplay of covariates sometimes diminishes or accentuates associations, which can be confusing. Consider clarifying these findings.

Thank you for this comment. We agree that the interplay among covariates can influence the observed associations, sometimes attenuating or accentuating effects. To clarify this, we have revised the Results and Discussion sections to better explain how covariate adjustment—particularly for age and sociodemographic variables—impacts the biomarker-cognition associations. This helps contextualize the observed changes across models.

DISCUSSION

- “GFAP is a sensitive biomarker for detecting and tracking A β pathology at early stages of AD” is likely an understatement. Multiple recent studies strongly support GFAP’s role in AD pathophysiology.

Thank you for this important observation. We have revised the manuscript to more accurately reflect the current evidence regarding GFAP. Rather than characterizing GFAP as merely a “sensitive biomarker,” we now acknowledge its increasingly recognized role in Alzheimer’s disease pathophysiology.

- This sentence is quite unclear as you do not state a direction of the associations: “Previously, the Health and Aging Brain among Latino Elders (HABLE) study also reported that higher levels of NfL were associated with several cognitive abilities, including processing speed, attention, executive function, and memory when covarying for age, sex, and education in a cohort that included Mexican-Americans”.

Thank you for pointing this out. We have revised the sentence for clarity and to reflect the direction of the association. It now reads:

“Previously, the Health and Aging Brain among Latino Elders (HABLE) study reported that higher levels of NfL were associated with poorer performance in several cognitive domains, including processing speed, attention, executive function, and memory, after adjusting for age, sex, and education in a cohort that included Mexican Americans.”

- You note that including individuals who are cognitively unimpaired and those with mild impairment could affect biomarker–cognition relationships. Why not stratify or at least discuss this distinction more explicitly?

Thank you for your comment. We agree that the inclusion of both cognitively unimpaired and mildly impaired individuals could influence the observed biomarker–cognition relationships. We conducted stratified analyses by cognitive status (cognitively unimpaired vs. mild cognitive impairment), which are now included in the Results and Discussion sections. These stratified findings help clarify the consistency and potential variability of associations across clinical subgroups.

FIGURES AND TABLES

- Table 1: Indicate total n for each category.

Thank you for the suggestion. We have revised Table 1 to report the sample size and percentages for each category.

- Table 2 is duplicated; also, clarify which cognitive domains the tests represent.

Thank you for catching the duplication. We have removed the redundant version of Table 2. Additionally, we have revised the table caption to clarify which cognitive domains each test represents, and we now briefly summarize this in the Methods section as well for improved clarity.

- With a large dataset, distributions of plasma biomarker values or composite cognitive scores might be quite insightful.

Thank you for this valuable suggestion. In response, we have included supplementary figures showing the distributions of plasma biomarker values and global cognitive composite scores. These visualizations provide additional context for interpreting the variability and central tendencies of these measures in our sample. We believe this strengthens the overall interpretation of the results.

- Consider adding line numbers for easier review.

Thank you for the suggestion. We have updated the manuscript to include line numbers in the tables and figures where applicable, to facilitate easier review and referencing.

- In the supplemental figure’s flow chart, clarify any points I noted in the attachment.

Thank you for reviewing the supplemental figure. We have carefully revised the flow chart to address the points you noted in the attachment.

SUPPLEMENTAL FIGURE

It is always helpful to have a flowchart, thank you for providing this. However, please consider providing the following:

- Abbreviations do not appear in alphabetical order in accompanying text

Thank you for your observation. We have revised the list of abbreviations in the manuscript to ensure that they now appear in alphabetical order in the accompanying text, in accordance with journal guidelines.

- Biospecimen collection = blood? (because it could be a myriad of things)

Thank you for your helpful suggestion. To improve clarity, we have revised the manuscript to replace the term "biospecimen collection" with "blood sample collection" to specify the sample type used for plasma biomarker analysis.

- Which covariates? And which covariates are missing?

Of the original sample, 496 participants were excluded due to missing data on one or more of these covariates. We moved this to the Covariates section to improve clarity. We have updated Supplementary Figure 1 to specify the number of missing observations for each covariate to increase transparency.

- What method was employed for removal of outliers (extreme values based on what)?

Thank you for your question. In the final version of the manuscript, the primary analyses now use natural log-transformed biomarker concentrations to address skewness and reduce the impact of extreme values. We retained the sensitivity analysis excluding values $\geq \pm 3$ SD from the mean to support consistency of findings.

- And also, were the outliers above or below the mean? For example, I can imagine NfL having high outliers whereas GFAP would have low outliers

We appreciate the opportunity to clarify. The primary analysis now uses natural log-transformed biomarker values, which reduces skew and minimizes the influence of extreme observations. However, we retained the outlier-exclusion approach ($> \pm 3$ SD from the mean) as a sensitivity check, and it yielded similar findings to the log-transformed models, supporting the robustness of our results. This is now detailed in both the Methods and Results sections, and illustrated in the Supplementary Table.

I wish you success with these revisions.

Thank you.

Reviewer #6 (Remarks to the Author):

This study addresses an important and timely topic by examining plasma biomarkers and cognitive performance among Hispanic/Latino adults—an underrepresented population in Alzheimer's disease research. The large sample size is a notable strength and offers a valuable opportunity to advance our understanding of blood biomarker utility within this group. However, several critical issues limit the current manuscript's contribution. Addressing these concerns would significantly strengthen the scientific rigor and relevance of the study.

1. While the authors highlight the underrepresentation of Hispanic/Latino populations, the manuscript does not sufficiently situate its findings within the broader scientific literature. A deeper comparison would help clarify the novelty of the results and determine whether the observed associations are consistent across populations.

Thank you for this important suggestion. We have expanded the Discussion section to better situate our findings within the broader literature. Specifically, we now compare our results to prior studies in non-Hispanic White, African American, and Hispanic/Latino populations (e.g., WHICAP, HABLE, HABS-HD), highlighting consistencies and differences in the associations between plasma biomarkers and cognitive outcomes. This addition helps clarify the novelty and generalizability of our findings.

2. The cognitive battery (B-SEVLT, Word Fluency, DSS) covers limited cognitive domains. While these tests offer valuable information, they do not represent "global cognition" as they do not fully capture global cognitive domains. Expanding the cognitive assessment to include additional domains—such as attention and visuospatial skills—would provide a more comprehensive understanding of cognition in this population.

Thank you for your comment. We acknowledge that our cognitive battery focuses on a limited set of domains. However, the composite score used in this study is conceptually similar to the Preclinical Alzheimer Cognitive Composite (PACC), which has been widely adopted in Alzheimer's disease (AD) research to capture subtle cognitive changes. Moreover, the term "global cognition" aligns with terminology used in prior HCHS/SOL and SOL-INCA studies, allowing consistency and comparability across publications. Nonetheless, we recognize that the battery does not assess all cognitive domains (e.g., visuospatial or attentional functions), and we now clarify this point in the Discussion as a limitation.

3. While CKD and BMI were included as covariates in the regression models, their direct relationships with biomarker levels were not fully explored. Given the distinct health profiles within this population, it would be informative to investigate how these covariates directly influence biomarker levels and cognitive outcomes. This could include subgroup analyses or interaction effects, which would provide deeper insights into these relationships.

Thank you for this insightful comment. We agree that chronic kidney disease (CKD) and body mass index (BMI) are important health factors that may influence both biomarker levels and cognitive outcomes. While our primary aim was to evaluate the associations between plasma biomarkers and cognition, CKD and BMI were included as covariates in the regression models to account for their potential confounding effects. Given the complexity of these relationships and the scope of the current study, we did not perform

subgroup or interaction analyses. However, we acknowledge this as an important area for future research and have noted it in the Discussion.

4. The finding that higher AD plasma biomarkers are associated with worse cognitive performance is expected. However, given the inclusion of both cognitively unimpaired and MCI individuals, stratified analyses are essential to provide a more nuanced understanding of these associations. Moreover, since the authors identify factors such as CKD and CVD risk as important covariates, stratifying analyses by these variables—alongside cognitive status—would offer deeper insights into how these health conditions may modify biomarker-cognition relationships.

Thank you for this suggestion. We have conducted stratified analyses by both age group (<60, 60–69, ≥70 years) and cognitive status (cognitively unimpaired vs. mild cognitive impairment), which are now included in the Supplementary Materials and referenced in the Results and Discussion. These analyses provide a more nuanced understanding of how biomarker-cognition associations may vary across different subgroups. While we recognize the value of also stratifying by CKD and cardiovascular risk, we have noted this as an important direction for future research.

5. Some sections of the discussion include content that is less relevant to the core findings. Focusing more on potential biological mechanisms—especially regarding why biomarker associations were attenuated after adjusting for CKD and CVD risk factors—would strengthen the discussion. For instance, elaborating on how vascular or metabolic health may influence biomarker-cognition associations could add meaningful context. Furthermore, the discussion should integrate relevant literature on racial/ethnic differences in biomarker profiles and cognition to enhance the impact of the findings.

Thank you for this constructive and insightful comment. In response, we have revised the Discussion to improve its alignment with the core findings of the study and to better contextualize the results within existing literature. Specifically:

- We streamlined the Discussion to reduce tangential content and ensure a tighter focus on our primary aims and results.
- We expanded on potential biological mechanisms, particularly how vascular and metabolic health (e.g., CKD, hypertension, diabetes) may contribute to elevated plasma biomarkers such as NfL and GFAP, and how these processes may in turn impact cognitive performance. This provides a plausible explanation for the attenuation of associations after adjusting for cardiometabolic risk factors and CKD status.
- We also incorporated relevant studies on racial/ethnic differences in plasma biomarker profiles and cognitive outcomes, including evidence showing variability in biomarker expression across ancestries, and how cardiometabolic disease burden—more prevalent in some Hispanic/Latino subgroups—may mediate or moderate these relationships.

These additions help to clarify the potential pathways linking biomarker levels to cognitive health in Hispanic/Latino populations and strengthen the relevance of our findings in the context of ADRD disparities and personalized medicine.

We appreciate the reviewer's recommendation, which significantly improved the clarity and impact of the Discussion section.

Reviewers' comments:

Reviewer #1 (Remarks to the Author):

Authors have revised all points appropriately. I appreciate the effort from the authors. I think this manuscript has enough quality for publication in Communications Medicine, after correcting the following minor point.

Page 6

"Blinded duplicated showed coefficients of variation <11.3%. We included ..., demonstrated coefficients of variation (CVs) below 12% across plates." -> "Blinded duplicated showed coefficients of variation (CV) <11.3%. We included ..., demonstrated CVs below 12% across plates."

We thank the reviewer for their supportive comments and careful reading. We have revised the text on page 6 as suggested to ensure consistency in terminology. The sentence now reads:

"Blinded duplicates showed coefficients of variation (CV) <11.3%. We included ..., demonstrated CVs below 12% across plates."

Reviewer #2 (Remarks to the Author):

The authors have done a great job addressing the long list of comments, and the manuscript has certainly improved a lot. However, not everything has been correctly addressed.

1. I suggested correcting for multiple comparisons per cognitive test per model, as many models are being executed here. The authors explained how they chose not to formally apply corrections here as biomarkers were selected a priori and highlight cautious interpretation in the limitations. However, I disagree. Even if variables were selected based on strong rationale, a lot of different tests are still being executed which gives a higher chance of a type I error occurring, something that still applies regardless of a priori decisions. Please add a column, or row, containing p-values corrected for multiple comparisons, to your Tables 2-4.

We appreciate the reviewer's thoughtful feedback and acknowledge the concern regarding inflated Type I error due to multiple testing. Since our biomarker selections were based on a priori hypotheses, we initially reported unadjusted p-values to directly address our primary research questions. However, in response to the concern, we have derived false discovery rate (FDR)-adjusted p-values to account for multiple comparisons within each cognitive measure per model (Supplemental Table 4). These adjusted p-values are presented in a new column. We have also updated the Methods section accordingly to reflect this addition and revised the Discussion to interpret the findings considering these corrected values.

2. My comments to add a M0 has been misinterpreted. I asked to show the effects of just age, sex, education and Hispanic/Latino background on cognition. My apologies if I was not clear enough. That would look like: cognition ~ age + sex + education + background. So no biomarkers included in the model. The results could be reported in a supplementary table for the main analyses only. Considering we know these sort of demographic factors have a substantial impact cognition, seeing their effects would improve our understanding of how much additional value biomarkers hold in addition to these demographic factors.

We thank the reviewer for the helpful clarification. We have added a new supplementary table (Supplementary Table 2) presenting results from regression models of cognitive outcomes on age, sex, education, and Hispanic/Latino background to establish a demographic baseline for cognitive performance. We view this addition as providing important descriptive context for understanding how demographic factors relate to cognitive performance, and for interpreting the added value of biomarker associations in the main models. While our objective was to examine associations between plasma biomarkers and cognitive performance, our primary analysis presents fitted regression models with plasma biomarkers as the exposure and adjusting for demographics, which also tests how much additional value biomarkers hold in addition to these demographic factors. We appreciate the reviewer's suggestion to enhance the clarity and interpretability of our findings and believe this addition strengthens the manuscript.

Reviewer #3 (Remarks to the Author):

Manuscript#: COMMSMED-25-0342A

I thank the Authors for addressing my concerns. I have some additional comments.

Major comment:

1. The interchangeable use of the terms AD and ADRD when referring to the plasma biomarkers is a source of confusion. This issue is apparent throughout the manuscript. In the Background section, it was stated that NfL and GFAP are non-specific biomarkers of AD and other brain diseases (which brain diseases? ADRD?). However, in the Discussion section the Authors referred to NfL and GFAP as biomarkers associated with AD pathophysiology (see lines 371-372 for an example). In the Background section, it would be useful to clarify which biomarkers were indicative of AD pathophysiology and which others were best associated with ADRD. The discussion of the results should be consistent with this understanding.

We thank the reviewer for this important observation regarding inconsistent terminology. We have revised the manuscript to clearly distinguish between biomarkers indicative of Alzheimer's disease (AD) pathophysiology and those that are non-specific markers of neurodegeneration or neuroinflammation more broadly associated with Alzheimer's disease and related dementias (ADRD). Specifically, we:

- Clarified in the Background section that plasma A β 42/40 and p-tau181 are considered biomarkers of AD pathophysiology, reflecting amyloid and tau-related processes, respectively.
- Noted that NfL is a marker of axonal injury and GFAP of astroglial activation, both of which are non-specific and can be elevated in a range of neurodegenerative conditions, including but not limited to AD.
- Replaced ambiguous uses of "AD" or "ADRD" throughout the manuscript with more precise language depending on the context of the biomarker and disease process being described.
- Updated the Discussion section (e.g., lines 371–372) to ensure consistency with these clarifications, and to avoid over-attribution of NfL and GFAP to AD-specific pathology.

We believe these revisions improve clarity and scientific accuracy in our characterization of the biomarkers and their relevance to AD and broader neurodegenerative processes.

Other comments:

2. Page 4, line 58: please specify that previous research refers to non-Hispanic White populations and add pertinent references.

Thank you for this suggestion. We clarified that most prior research has been conducted in predominantly non-Hispanic White populations and have added appropriate references to support this point.

3. page 4, line 62: please specify that age and cognitive status were also taken into account.

We have revised the sentence to explicitly state that age and cognitive status were considered in evaluating the consistency of biomarker-cognition associations in this population.

4. A brief explanation on the criteria used to assign a diagnosis of MCI should be added in the paragraph 'Neuropsychological assessment'. The title of the paragraph could be changed into 'Neuropsychological assessment and cognitive status'.

We thank the reviewer for this helpful suggestion. We have revised the section title to "Neuropsychological Assessment and Cognitive Status" and added a brief explanation of the Mild Cognitive Impairment (MCI) diagnostic criteria used in SOL-INCA directly within this section (Page 7). Classification was based on neuropsychological test performance in the impaired range (≥ 1 SD below the mean) on any cognitive test (memory, executive function, language, or processing speed) relative to the SOL-INCA robust internal norms adjusting for age, sex, education, and language-adjusted mean and Picture Vocabulary Test; evidence of global cognitive decline from baseline exceeding -0.055 SDs per year; self-reported cognitive decline on the brief Everyday Cognition Scale (ECog-12); no or minimal functional impairment in instrumental activities of daily living. Participants with suspected severe cognitive impairment were excluded.

5. Supplementary Figure 1 should report a brief explanation for the terms 'missing exposure' and 'censored'. The same terms should also be referred to in the text (pag.6, lines 110-113).

We thank the reviewer for this helpful suggestion. We have updated the legend of Supplementary Figure 1 to define 'missing exposure' as the absence of plasma biomarker data due to unavailable blood samples or

assay failure, and 'censored' as observations falling outside the limits of detection for the assay. These terms have also been defined in the main text on page 6, lines 110–113, to enhance clarity.

6. Pag.8: the sentence 'we interpreted findings cautiously in the context of multiple testing' is unclear. Please specify.

We thank the reviewer for pointing this out. We have revised the sentence to clearly state our approach to multiple comparisons. Specifically, we now clarify that although formal correction was not applied in the primary analysis due to the hypothesis-driven nature of the study, we conducted a sensitivity analysis using the Benjamini-Hochberg false discovery rate (FDR) method to account for multiple testing within each cognitive measure.

7. The number of MCI should be reported in the paragraph 'Target population characteristics.'

We thank the reviewer for this suggestion. We have added the number of participants classified with mild cognitive impairment (MCI) to the "Target population characteristics" paragraph to provide a clearer description of the sample's cognitive status.

8. Table 1: the second and fourth lines [Mean (SD), N (%)] may be eliminated. Instead, the sentence 'Values denote mean(SD) or number(%)' may be added to the notes. As regards hypertension, dyslipidemia, and CKD status, only the values referred to the presence of the conditions should be reported. APOE4: 'negative' should be 'non carriers'.

We thank the reviewer for these helpful formatting and terminology suggestions. We have made the following changes to Table 1:

- Removed the second and fourth lines indicating "Mean (SD)" and "N (%)" from the table body.
- Added the sentence "Values denote mean (SD) or number (%)" to the table footnote for clarity.
- Updated the values for hypertension, dyslipidemia, and CKD status to report only the number and percentage of participants with the condition.
- Replaced "APOE4 negative" with "non-carriers" to ensure accurate and appropriate terminology.

9. Table 3 and 4 overlap with Figure 2 and 3. I suggest to move them into the supplemental material.

We thank the reviewer for this helpful suggestion. In response, we have moved Tables 3 and 4 to the supplementary materials as Supplementary Tables 5 and 7, respectively, to reduce redundancy with Figures 2 and 3. The main text has been updated to reflect this change.

Reviewer #4 (Remarks to the Author):

To the kind attention of Dr.Marquez et al.,

I would like to congratulate you with the revision work of the manuscript, especially considering you received five reviewer reports and you had addressed all comments very thoroughly. The comments I provided were all addressed with adjusting the text, new statistical analyses and new figures. Therefore I am satisfied with the updated version of the manuscript. The added insight provided by the stratification of patients and added figures have elevated the insights provided by the manuscript.

We sincerely thank the reviewer for their kind words and thoughtful evaluation of our revised manuscript. We greatly appreciate your constructive feedback throughout the review process, which helped us to strengthen the clarity, rigor, and overall impact of the work. We are pleased to hear that the stratified analyses and added figures provided valuable insight, and we thank you again for your support in improving the manuscript.

Reviewer #5 (Remarks to the Author):

Dear Dr. Marquez,

Thank you for your thorough revisions. I appreciate your diligent efforts in addressing the points raised in my review. I am satisfied with the changes made to the manuscript, particularly the improvements in clarity, methodological detail, and the discussion of key findings.

While a few minor points of discrepancy remain, they do not detract from the overall quality and enhanced rigor of the work. I believe the manuscript has been significantly strengthened and is now well-positioned for publication.

We sincerely thank the reviewer for their thoughtful feedback and encouraging comments. We are pleased that the revisions improved the clarity, methodological transparency, and interpretation of findings. Your insights have been instrumental in strengthening the manuscript, and we are grateful for your time and contributions throughout the review process.

Reviewer #6 (Remarks to the Author):

The revised manuscript provides a much-improved contextualization of the findings, with a clear comparison to prior studies in non-Hispanic White, African-American, and Hispanic/Latino cohorts. This addition effectively highlights both the novelty and consistency of your results. Your justification for the cognitive composite—positioned as analogous to the PACC—and the acknowledgment of its limited domain coverage are appropriate and adequately address my earlier concern. Retaining CKD and BMI as a priori covariates, while reserving interaction analyses for future work, is a reasonable decision given the cross-sectional nature of the study. Most notably, the addition of stratified analyses by age (<60, 60–69, ≥70 years) and cognitive status (CU vs. MCI), with results clearly presented and referenced, fully addresses my request regarding potential effect heterogeneity. I appreciate the thoughtful revisions and have no further comments.

We sincerely thank the reviewer for their detailed and thoughtful assessment of the revised manuscript. We are especially grateful for your recognition of the efforts made to strengthen contextualization, clarify methodological choices, and address effect heterogeneity through stratified analyses. Your feedback has been invaluable in enhancing the rigor and clarity of the manuscript, and we greatly appreciate your support throughout the review process.